# CONTEXTUALIZED GENERATIVE RETRIEVAL

## ABSTRACT

The text retrieval task is mainly performed in two ways: the bi-encoder approach and the generative approach. The bi-encoder approach maps the document and query embeddings to common vector space and performs a nearest neighbor search. It stably shows high performance and efficiency across different domains but has an embedding space bottleneck as it interacts in L2 or inner product space. The generative retrieval model retrieves by generating a target sequence and overcomes the embedding space bottleneck by interacting in the parametric space. However, it fails to retrieve the information it has not seen during the training process as it depends solely on the information encoded in its own model parameters. To leverage the advantages of both approaches, we propose Contextualized Generative Retrieval model, which uses contextualized embeddings (output embeddings of a language model encoder) as vocab embeddings at the decoding step of generative retrieval. The model uses information encoded in both the non-parametric space of contextualized token embeddings and the parametric space of the generative retrieval model. Our approach of generative retrieval with contextualized vocab embeddings shows higher performance than generative retrieval with only vanilla vocab embeddings in the document retrieval task, an average of 6% and 18% (25%) higher performance in KILT (NQ, TQA) R-precision and NQ-320k Hits@1 (@10), respectively, suggesting the benefits of using contextualized embedding in generative retrieval models.[1]

## 1 INTRODUCTION

Text retrieval is often formulated as finding the most relevant items from a large corpus given an input query. The bi-encoder approach of using an encoder to map the documents and the query to a common vector space and performing a nearest neighbor search has been a common practice in text retrieval tasks (Karpukhin et al., 2020; Wu et al., 2020; Ni et al., 2021). Despite its high performance and popularity, it has an embedding space bottleneck (Luan et al., 2021; Lee et al., 2022; Cao et al., 2021). The performance decreases as the document length increases due to the limited expressiveness of fixed-size document embeddings. Also, it misses the fine-grained interaction between the query and the document as they interact in L2 or inner product space. The bi-encoder approach also requires large storage space to save all document embeddings.

A recently-proposed alternative to the bi-encoder approach is using a generative retrieval model (Cao et al., 2021; Tay et al., 2022; Bevilacqua et al., 2022; Lee et al., 2022) which retrieves the most relevant sequence by generating the item token-by-token, where the item is the identifier of the target sequence or the sequence itself (e.g., title, passage, document ID). They show high performance while using a low storage footprint by overcoming the embedding space bottleneck. These models interact in the parametric space of the language model rather than just in the inner product space. However, as existing generative retrieval models rely solely on the information encoded in their own parameters, the model cannot retrieve the correct target sequence if it has not seen such information during the training process.

---

[1]We will make our code publicly available.

To this end, we propose *contextualized* generative retrieval model (CGR), a retrieval model that overcomes the aforementioned limitations of existing generative retrieval models by leveraging contextualized vocab embeddings (output embeddings of language model encoder) to make use of non-parametric information from the context surrounding the vocab tokens. It uses not only the parametric space of the model as in generative retrieval models but also the non-parametric space of contextualized target embeddings (external memory) as in bi-encoder models. As in Figure 1, the model has two submodules: (1) an EMBedding model (EMB), which is an encoder model that outputs contextualized embeddings, and (2) a RETrieval model (RET), which is an encoder-decoder model that retrieves a target sequence when given an input query. The model first constructs the contextualized embedding matrix with the output embeddings of EMB and uses the matrix as the decoder vocab embeddings when training RET. By utilizing the contextualized embedding matrix rather than the vanilla embedding matrix while generating a target sequence, RET uses both information encoded in its own parameters as existing generative retrieval models and information encoded in the contextualized embeddings. Also, as RET uses the contextualized embeddings during both the training and inference step, RET is optimized to utilize the information encoded in the contextualized embeddings.

We show the importance of using external memory (non-parametric space) of contextualized target embedding in generative retrieval models by comparing the performance between CGR and GENRE (Cao et al., 2021), a generative retrieval model which only operates on the parametric space. CGR shows an average of 6% increment in Natural Questions (Kwiatkowski et al., 2019) and TriviaQA (Joshi et al., 2017) in KILT (Petroni et al., 2021) and 18% (25%) higher performance in Hit@1 (@10) of NQ-320k. We also compare the results with different baselines for a comprehensive understanding of the model performance.

The main contributions of our paper are as follows:

- We present Contextualized Generative Retrieval (CGR), a generative retrieval model which uses the contextualized embedding matrix while generating a target sequence. It shows an average of 6% and 18% (25%) higher performance in KILT (NQ, TQA) R-precision and NQ-320k Hits@1 (@10), respectively, compared to GENRE in the same setting.
- We show that using contrastive learning as intermediate training further increases the performance of the contextualized generative retrieval model by a large margin.
- We perform extensive ablation studies and analysis over several variants of contextualized generative retrieval models for a comprehensive understanding of how to use contextualized embeddings and why using contextualized embeddings is better than using vanilla vocab embeddings.

## 2 RELATED WORK

**Generative Retrieval** Existing generative retrieval models retrieve relevant items by generating either the identifiers or entire sequences of the items. Cao et al. (2021) propose GENRE (Generative ENtity REtrieval), which retrieves a document by generating the titles with constrained beam search. Tay et al. (2022) propose DSI (Differentiable Search Index), which assigns a unique ID to each item in the corpus and trains the model to encode all information of the document and the ID in the model parameters. During the inference step, DSI generates the ID of the most relevant document. Wang et al. (2022) propose NCI (Neural Corpus Indexer), which also retrieves by generating the document ID as in DSI, but improves performance by query generation and prefix-aware weight-adaptive decoder. Bevilacqua et al. (2022) propose SEAL (Search Engines with Autoregressive LMs), which can retrieve any span from any position in the corpus by using the compressed full-text substring index (FM-Index). In this work, we propose Contextualized Generative Retrieval which generates the target sequence by utilizing the contextualized embedding matrix rather than the vanilla vocab embedding matrix as in the aforementioned generative retrieval models. Therefore, the model utilizes both the parametric space of the generative retrieval and the non-parametric space of contextualized token embeddings. To the best of our knowledge, we are the first to utilize the contextualized token embeddings on generative retrieval models.

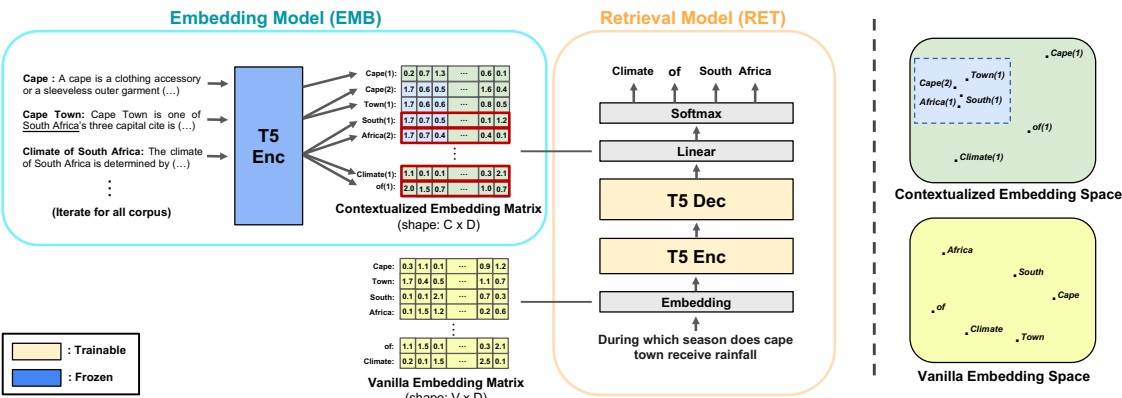

Figure 1: The left side shows the model architecture of CGR$_{\text{Base}}$, and the right side shows the difference between contextualized embedding space and vanilla embedding space which is constructed with contextualized embedding matrix and vanilla embedding matrix, respectively. Contextualized embedding matrix is constructed by the output embeddings from EMB and is used as the decoder vocab embeddings in RET; existing generative retrieval utilizes the vanilla embedding matrix when generating the target sequence, RET utilizes the contextualized embedding matrix. In the document retrieval task with a title as the target sequence, the title of the document and its corresponding content from the document is given as input to EMB and we only save the output embeddings of the title.

**Retrieval Models with Contextualized Token Embedding** ME-BERT (Luan et al., 2021),Col-BERT (Khattab & Zaharia, 2020) and COIL (Gao et al., 2021a) are retrieval models which retrieve the target sequence by utilizing the multiple contextualized token embeddings. It has shown high performance by leveraging the benefits of the cross-encoder architecture in bi-encoder architecture. CGR also utilizes the contextualized token embeddings, but differs from the three models in that while they interact in the inner product space, CGR has the benefit of interacting in the parametric space.

**Semi-Parametric Models** KNN-LM (Khandelwal et al., 2020), RAG (Lewis et al., 2020) and RETRO (Borgeaud et al., 2022) are semi-parametric models which use both the parametric space of the model and the non-parametric space. KNN-LM improves the LM performance by generating the next token during the inference step by interpolating between the nearest neighbor distribution (distance in the contextualized embedding space) and the model vocab distribution. RAG and RETRO are semi-parametric models that first retrieve relevant texts with the retriever in a non-parametric manner and generate the output based on the retrieved texts. CGR also utilizes both the parametric and non-parametric space as the three models do. However, it differs from KNN-LM in that it is trainable and from RAG and RETRO in that CGR uses the non-parametric space for the decoder vocab embeddings.

## 3 CONTEXTUALIZED GENERATIVE RETRIEVAL

Generative retrieval is the task of retrieving the most relevant retrieval target (e.g., title, passage, document identifier) by generating the retrieval target token-by-token when given an input query. The training objective of the generative retrieval model is to maximize

$$P((t_1, \cdots, t_n)|q) \propto \prod_{i=1}^{n} P(t_i|q, t_{<i}) \tag{1}$$

where $t_*$ denote the tokens of the retrieval target. Such an approach has shown high performance while using a low storage footprint (Cao et al., 2021; Tay et al., 2022; Bevilacqua et al., 2022; Lee et al., 2022). However, it has limitations in that the model depends solely on the information encoded in its own parameters. Thus,

the model fails to retrieve the correct target sequence if it has not seen the information during the training process.

To overcome the limitations, we propose *Contextualized* Generative Retrieval model (CGR), a generative retrieval model which uses not only the parametric space of the model but also the non-parametric space (external memory) of contextualized token embeddings[2] to leverage the benefits of the bi-encoder model and combine the advantages of the two models. CGR (Figure 1) utilizes the contextualized embeddings (output embeddings of the language model encoder) rather than the vanilla vocab embeddings while generating the retrieval target. Therefore, the model does not depend only on the information encoded in its own parameters, but can also take advantage of non-parametric information during generation as CGR encodes the document content into contextualized vocab embeddings. Also, by allowing a single token to have multiple token embeddings, the model can learn about the different meanings of the token and the different contexts in which it is used. Therefore, the embedding space constructed with multiple contextualized token embeddings (contextualized embedding space) will become more expressive and fine-grained than the embedding space constructed with model vocab embeddings (vanilla embedding space).

CGR differs from existing generative retrieval models by which token embedding matrix is used for the decoder model. Generative retrieval models such as GENRE (Cao et al., 2021) utilize the pre-trained language model architecture as-is: both the encoder and the decoder model share the same vanilla vocab embedding matrix of shape $V \times D$ where $V$ is the vocab size, and $D$ is vocab embedding dimension. CGR whereas uses different vocab embedding matrices for the encoder and decoder model: the encoder model uses the vanilla vocab embedding matrix ($V \times D$) as in existing generative retrieval models, but the decoder model uses the contextualized embedding matrix of shape $C \times D$ where $C$ is the number of contextualized embeddings. $C$ is larger than $V$ as a token is matched with multiple contextualized embeddings in most cases (e.g., in Figure 1, same token "Cape" has two different contextualized embeddings which we name as Cape(1) and Cape(2) to differentiate the two.), but $C$ can be reduced with practical tactics (Section 4.4).

CGR is composed of two submodules: (1) an EMBedding model (EMB), which is an encoder model that outputs meaningful contextualized embeddings, and (2) a RETrieval model (RET) which is an encoder-decoder model that retrieves a target sequence by generating the sequence while utilizing the information encoded in the contextualized embedding matrix. For example, when the retrieval target sequence is the title of a document, we pass the concatenation of the title and its corresponding document content as the input to EMB and save the pair of input tokens and their output embeddings. By passing not only the title but also its corresponding document content to the model, the output embeddings contain both the information of the title and the document content. In practice, for efficiency, we only sample a few embeddings that are deemed to be the most informative and representative of the target sequence, or simply the first few embeddings. The extracted contextualized embeddings are then used to form the contextualized embedding matrix, which serves as the decoder vocab embeddings of RET. As the vocab embedding matrix of the encoder and decoder model have different shapes, we assign different token IDs for the decoder; a unique token is paired with multiple token IDs where each ID indicates different contextualized embedding in the decoder.

## 4 MODEL DETAILS

In Section 3, we show the overall architecture of CGR that can be applied to general tasks. In this section, we present details of how we design CGR for practical usage in document retrieval tasks with the document title as the retrieval target. The ideal design of CGR is to use the encoder of RET as EMB for every gradient update during the training step to ensure the high coherency between the contextualized embeddings and RET. However, such a method requires high computational cost as it needs to construct contextualized embedding matrix at every step. Therefore, we present practical models; the base architecture of contextualized generative retrieval (CGR$_{Base}$), and two improvements, CGR$_{Async}$ and CGR$_{Contra}$. Also, we show how

---

[2]In this paper, contextualized embeddings refer to the output of language model encoder, which can incorporate information from the nearby context.

we reduce the number of contextualized embeddings. We add the figure of each model and more details in Appendix A.

## 4.1 CGR$_{\text{BASE}}$

Base CGR (CGR$_{\text{Base}}$) in Figure 1 is the most basic contextualized generative retrieval model among the ones we propose. It uses the pre-trained T5 encoder as EMB and the T5 encoder-decoder as RET. EMB is frozen during the training step, and only RET is trainable.

## 4.2 CGR$_{\text{ASYNC}}$

Asynchronous CGR (CGR$_{\text{Async}}$) is a model where EMB is *asynchronously replaced* by the encoder of RET for every $N$ epochs. When the model parameters of EMB are replaced, we construct a new contextualized embedding matrix with the replaced EMB and resume the training. As the decoder vocab embeddings of RET are updated every $N$ epoch, EMB and RET of CGR$_{\text{Async}}$ would have more coherency between each other compared to CGR$_{\text{Base}}$. We keep $N = 20$ for all experiments. See Appendix C.1 for details on how $N$ affects the performance.

## 4.3 CGR$_{\text{CONTRA}}$

Bi-encoder retrieval models with contrastive loss have shown high performance, as the model learns and constructs well-structured global embedding space and regularizes the space to be uniform (Ni et al., 2021; Gao et al., 2021b; Gao & Callan, 2022; Izacard et al., 2022). CGR with Contrastive Learning (CGR$_{\text{Contra}}$) is designed to leverage such benefits of contrastive learning in a contextualized generative retrieval model; the model is first trained with contrastive loss and then on the generative retrieval objective, i.e., retrieving the most relevant sequence by generating the sequence token-by-token.

**Step 1. Token-level Contrastive Learning**   We first train RET with token-level contrastive loss to allow the model to learn the overall search space of token embeddings in the target corpus. Given a training dataset of pairs $\{(\boldsymbol{q}, \boldsymbol{t})\}$ where $\boldsymbol{q}$ is the query text, and $\boldsymbol{t}$ is the retrieval target (title of the document to retrieve) composed of multiple tokens $\boldsymbol{t}_i$ ($1 \leq i \leq k$ where $k$ is the length of the target), we split the dataset into $k$ separate pairs $\{(\boldsymbol{q}, \boldsymbol{t}_i)\}$ where $\boldsymbol{t}_i$ is a token of $\boldsymbol{t}$ to construct the training dataset of query-token. With the query-token dataset, we train the first output token embedding from the decoder of RET to be close to all token embeddings in $\mathcal{T}^+$ when given query $\boldsymbol{q}$ as an input to RET. $\mathcal{T}^+ = \{\mathbf{t}_1^+, \cdots, \mathbf{t}_k^+\}$ ($k = |\mathcal{T}^+|$) is a set of positive token embeddings (tokens that make up one title), and $\mathcal{T}^- = \{\mathbf{t}_1^-, \cdots, \mathbf{t}_{|\mathcal{T}^-|}^-\}$ is the set of negative token embeddings, which are *all* other token embeddings in contextualized embedding matrix. The objective is to minimize the contrastive loss to make the query text embedding $\mathbf{q}$ be closer to all positive token embeddings in $\mathcal{T}^+$:

$$L(\mathbf{q}, \mathbf{t}_1^+, \cdots, \mathbf{t}_{|\mathcal{T}^+|}^+, \mathbf{t}_1^-, \cdots, \mathbf{t}_{|\mathcal{T}^-|}^-) = -\log \frac{\sum_{\mathbf{t}^+ \in \mathcal{T}^+} e^{<\mathbf{q}, \mathbf{t}^+>}}{\sum_{\mathbf{t}^+ \in \mathcal{T}^+} e^{<\mathbf{q}, \mathbf{t}^+>} + \sum_{\mathbf{t}^- \in \mathcal{T}^-} e^{<\mathbf{q}, \mathbf{t}^->}} \tag{2}$$

where $\langle \ , \ \rangle$ is the inner product value between the two embeddings. As we have the whole set of contextualized embeddings of the corpus (contextualized embedding matrix), it is possible to consider all other embeddings as negative. See Section C.2 for more details and analysis of different contrastive loss and model architecture.

**Step 2. Generative Retrieval**   We further train RET from step1 with a generative retrieval objective, which is the same as in CGR$_{\text{Base}}$. To be specific, we use the encoder of RET in step1 as EMB of step2 to extract the contextualized embeddings, and use the RET in step1 as the initial parameter of RET in step2.

| Training Dataset | Single ($\leq 3\%$) | | NQ+TQA ($\leq 5\%$) | | All KILT (100%) | |
|---|---|---|---|---|---|---|
| Model | NQ | TQA | NQ | TQA | NQ | TQA |
| GENRE | 51.8* | 65.0* | 52.7* | 64.8* | 60.3 | **69.2** |
| CGR$_{Base}$ | 59.0 | 68.2 | 59.4 | 68.7 | - | - |
| CGR$_{Async}$ | 59.2 | 68.4 | 59.8 | 68.7 | - | - |
| CGR$_{Contra}$ | 59.8 | **68.6** | **60.3** | **68.9** | - | - |
| BM25 | 23.4$^\dagger$ | 25.2$^\dagger$ | 23.4$^\dagger$ | 25.2$^\dagger$ | 23.4$^\dagger$ | 25.2$^\dagger$ |
| DPR | **60.1**$^\dagger$ | 63.9$^\dagger$ | 59.5$^\dagger$ | 62.9$^\dagger$ | 59.4 | 61.5 |
| SEAL | - | - | - | - | **63.2** | 68.4 |

Table 1: R-precision(%) for document retrieval task on NQ and TQA test dataset in KILT. Results except CGR are from the KILT leaderboard. The column of the table is divided by how many training datasets are used. Numbers in the bracket are the rate of the number of training datasets over the number of training datasets when using all KILT datasets. Results with * in GENRE are from GENRE*. Underlined model is direct baseline of CGR$_{Base}$. Results with $\dagger$ in BM25 and DPR are trained in same setting as CGR (Appendix B.2). Best in **bold.**

## 4.4 CLUSTERING

To construct the contextualized embedding matrix to be used as the vocab embedding matrix of the decoder of RET, we first extract all contextualized embeddings of each target token with EMB. As it requires a large storage footprint to save all the embeddings, we reduce the number of embeddings by using clustering and saving only the representative embeddings of each cluster. To be specific, we perform k-means clustering over the contextualized embeddings of the same token (which might have different surrounding contexts) and leave only the $k$ centroid embeddings[3] as the decoder vocab embeddings of the token. We keep $k = 5$ for all experiments. When $k = 5$, it only requires 0.3% of storage footprint compared to when saving all contextualized token embedding. Also, it requires 0.34GB more storage compared to the vanilla vocab embeddings ($k = 1$) which is marginal compared to the storage footprint to save the model parameters (3GB). See Appendix C.3 for examples, and how $k$ affects the performance and the storage footprint.

## 5 EXPERIMENTS

In Section 5.1, we describe the baselines, datasets, and basic setup used in our experiment. In Section 5.2, we show both qualitative and quantitative effectiveness of using contextualized embeddings by comparing CGR with baseline models. We also compare the performance and characteristics among the variants of CGR.

## 5.1 SETUP

We compare CGR with six baselines widely used in document retrieval task: BM25, GENRE, DSI, NCI, SEAL, and DPR. BM25 (Robertson & Zaragoza, 2009) is a term-matching model relying on an efficient algorithm. DPR (Karpukhin et al., 2020) is a bi-encoder retrieval model which retrieves the most relevant document by mapping a query and documents to a common vector space and performing a nearest neighbor search. See Section 2 for descriptions of GENRE, DSI, NCI, and SEAL. We compare all models using a document retrieval task, where the input is a query, and the output is a sequence related to relevant Wikipedia documents (e.g., title, document ID). See Appendix B for more details.

**KILT (NQ, TQA)** We train CGR with two datasets, Natural Questions (NQ) (Kwiatkowski et al., 2019) and TriviaQA (TQA) (Joshi et al., 2017) from the KILT dataset (Petroni et al., 2021), a benchmark for knowledge-intensive language tasks with eleven different datasets spanning five different tasks (fact checking, question answering, entity linking, dialogue, and slot filling). It gathers data in different formats into a

---

[3]When the number of extracted contextualized embeddings of a token is smaller than $k$, we do not perform k-means clustering but use its own contextualized embedding. Also, we use a single non-contextualized embedding for special tokens such as the EOS token or PAD token.

Table 2: Hits@1, Hits@10 in NQ-320k. Results of BM25&DSI are from Tay et al. (2022) and NCI&NCI- are from Wang et al. (2022). NCI- is NCI without query generation to match the number of training datasets with other models. All models are based on T5-large. Underlined model is direct baseline of CGR$_{Base}$. Best in **bold.**

| Training Dataset | Model | Hits@1 | Hits@10 |
|---|---|---|---|
| | BM25 | 11.6 | 34.4 |
| | DSI | 35.6 | 62.6 |
| NQ-320k | NCI- | 53.6 | 67.8 |
| | GENRE* | 53.7 | 64.7 |
| | CGR$_{Base}$ | 62.2 | 78.8 |
| | CGR$_{Contra}$ | **63.4** | **81.1** |
| NQ-320k + additional datasets | NCI | 88.7 | 95.8 |

Table 3: R-precision(%) for the document retrieval task on NQ and TQA test dataset in KILT. We compare the results of GENRE*, CGR$_{Base}$-title-only and CGR$_{Base}$ where the models are trained with NQ+TQA (Section 5.2). The results show the importance of extracting contextualized embeddings with not only the title but also the corresponding document content.

| | GENRE* | CGR$_{Base}$-title-only | CGR$_{Base}$ |
|---|---|---|---|
| NQ | 52.7 | 58.4 | **59.4** |
| Trivia | 64.8 | 68.2 | **68.7** |

common format, and the corresponding datasets share the same snapshot of Wikipedia as the corpus. DPR and GENRE are trained with all nine datasets in KILT[4]. Contextualized embeddings of the target retrieval sequence (title of the page) are the output embeddings from EMB when the title and its corresponding document content are given as the input. We evaluate all results with R-precision, a metric widely used to evaluate retrieval performance in KILT. It is calculated as $\frac{r}{R}$ where $R$ is the number of Wikipedia documents in each provenance set, and $r$ is the number of related documents among the top-$R$ retrieved documents.

**NQ-320k** To compare with Tay et al. (2022), we experiment on NQ-320k, a restricted setting from the official NQ dataset; it uses about 4% of Wikipedia corpus as the corpus set[5]. We construct the contextualized embedding matrix with the title of the document and its corresponding content as the input to EMB. The results are evaluated using Hits@N (N={1, 10}), which shows the proportion of the correct documents ranked in the top N predictions.

## 5.2 RESULTS

**KILT (NQ, TQA)** Results in Table 1 show that CGR$_{Contra}$ outperforms GENRE* by 6% which demonstrates the effectiveness of contextualized embeddings. For both cases where the model is trained over a single dataset and over NQ and TQA together (NQ+TQA), all CGR variants show higher performance over GENRE*. All CGR variants trained jointly on NQ and TQA show higher performance than those trained on a single dataset (NQ or TQA). Such results suggest that CGR tends to improve the performance when trained with more datasets. Note that due to limited available resources, we did not train CGR with the full KILT dataset (ALL KILT) as in GENRE[6], DPR, or SEAL. However, CGR$_{Contra}$ trained on less than 5% of the training dataset from the full KILT dataset show higher or comparable performance to those models. See Appendix C.7 for results in the KILT dev set.

**NQ-320k** We report the results on NQ320k in Table 2, which uses about 4% of Wikipedia corpus as the corpus set, to compare CGR with DSI and NCI which only experiment over NQ-320k. We compare the results between CGR$_{Base}$, CGR$_{Contra}$, and baselines (BM25, DSI, NCI, GENRE*). CGR$_{Contra}$ shows the highest performance when trained on the same number of datasets; 18% and 25% higher performance to GENRE* in Hits@1 and Hits@10, respectively. We also compare the result of CGR$_{Base}$ with DSI as a direct baseline in Appendix C.8.

---

[4]Due to limited resources, we did not train CGR with full KILT datasets as in GENRE, which used 128 V100 GPUs with 32GB of memory for about 33 hours. For a fair comparison with CGR, we train GENRE*, GENRE trained with the same resource, same pre-trained model (T5), hyperparameter, and dataset as CGR.

[5]The corpus set is the union of train/dev/test target sequences. As exact splits, document ID, and preprocessing code used by Tay et al. (2022) are not released, we tried to replicate the setting as closely as possible when constructing the NQ-320k dataset to train CGR$_{Base}$.

[6]GENRE uses 128 V100 GPUs with 32GB of memory for about 33 hours.

Table 4: R-precision(%) for the test sets of document retrieval tasks in KILT. Both GENRE* and CGR$_{Base}$ are trained with NQ + TQA; other datasets are not seen during the training time. Best in **Bold.**

|  | In-Domain Datasets | | Out-of-Domain Datasets (Inference Only, Zero-Shot) | | | | | | | | |
|  | NQ | TQA | FEVER | AY2 | WnWi | WnCw | T-REX | zsRE | HoPo | ELI5 | WoW |
|---|---|---|---|---|---|---|---|---|---|---|---|
| GENRE* | 52.7 | 64.8 | 64.2 | 9.1 | 2.8 | 3.4 | 53.9 | 76.1 | 34.3 | 11.2 | 48.9 |
| CGR$_{Base}$ | **59.4** | **68.7** | **67.0** | **10.3** | **5.4** | **7.8** | **59.1** | **79.2** | **37.5** | **12.5** | **51.7** |

**GENRE\* vs. CGR$_{Base}$ in Zero-Shot Setting**  Table 4 shows that CGR$_{Base}$ is stronger than GENRE* in the zero-shot setting where the models are trained on NQ and TQA and are evaluated on the other 9 datasets in KILT that are not used during the training step. CGR$_{Base}$ shows an average of 3% improvement from GENRE*. CGR$_{Base}$ shows high performance on information unseen during the training step as it does not solely rely on the information encoded in the parametric space (information that the model sees during the training step) but also on the non-parametric space of the contextualized embeddings.

**Differences among CGR$_{Base}$, CGR$_{Async}$, and CGR$_{Contra}$**  In Table 1, we can see that CGR$_{Contra}$ shows consistently higher performance than CGR$_{Base}$ and CGR$_{Async}$. We hypothesize two factors for such improvements. First, as the model is trained on contrastive learning before training on generative retrieval task, CGR$_{Contra}$ can leverage the benefits of contrastive learning where it learns and constructs well-structured overall embedding space and regularizes the space to be uniform (Ni et al., 2021; Gao et al., 2021a;b; Izacard et al., 2022). We check the quality of the embedding space with $L_{uniformity}$ proposed in Wang & Isola (2020), where the numbers represent how uniform the embedding space is. CGR$_{Contra}$ (-19.7) shows a lower number than CGR$_{Base}$ (-18.2) where the lower the better. Second, as EMB is initialized with the encoder of RET, there is high coherency between EMB and RET. The importance of having high coherency between EMB and RET can also be seen from the performance gain from CGR$_{Base}$ to CGR$_{Async}$; CGR$_{Base}$ uses the initial EMB without any replacement, but CGR$_{Async}$ replaces EMB with the encoder of RET every $N$ epochs and shows higher performance as $N$ decreases. i.e., the update is more frequent. More details in Appendix C.4.

**Importance of Having Contextualized Embeddings with Document Content**  In Table 3, we compare the results between CGR$_{Base}$-title-only, a model trained with contextualized embeddings extracted with *only the title* as the input to EMB, and CGR$_{Base}$, a model trained with contextualized embeddings extracted with *both* the title and corresponding document content as input to EMB. CGR$_{Base}$-title-only can be considered as an intermediate model between CGR$_{Base}$ and GENRE* as it uses the non-parametric space but is constructed with limited information (only with the title, without the entire document content). CGR$_{Base}$ shows the highest performance, GENRE* shows the lowest performance, and the performance of CGR$_{Base}$-title-only is in-between the two models, suggesting that there is a correlation between the performance and how much contextual information is in the non-parametric space.

GENRE*, which uses vanilla vocab embedding as the target embedding, has to depend solely on the information encoded in its own parameters (the parametric space of the generative retrieval model). On the other hand, CGR$_{Base}$ and CGR$_{Base}$-title-only can depend on not only the parametric space of the generative retrieval model as GENRE* does but also the non-parametric space of corpus information embedded in the contextualized target embedding. By utilizing the contextualized target embedding, the model can know in which context the token is used and discern documents with different contexts.

Although both CGR$_{Base}$ and CGR$_{Base}$-title-only utilize contextualized target embeddings, the contextualized target embedding of CGR$_{Base}$-title-only contains constrained information compared to that of CGR$_{Base}$. Therefore, CGR$_{Base}$-title-only fails on cases where the document content is necessary to retrieve the target sequence successfully. Table 5 shows examples where there is no direct relationship between the query and the target sequence such as lexical overlap or semantic similarity. It is difficult for the model to predict the target without the help of the document content about what information is in the document or what relationship exists between the query and the target sequence. We can see from the table (Table 5) that CGR$_{Base}$ successfully retrieves as such information is embedded in the contextualized target embeddings whereas

Table 5: Top-3 prediction results of CGR$_{Base}$, CGR$_{Base}$-title-only, and GENRE* on NQ dev set in KILT. Highlights on the correct target sequence.

| Query | Prediction Results | | | |
|---|---|---|---|---|
| what do the 3 dots mean in math | CGR$_{Base}$ | Therefore sign , Infinity symbol, Equation | | |
| | CGR$_{Base}$-title-only | Slashed zero, Homo sapiens, Equation | | |
| | GENRE | Ellipsis, Infinity symbol, Homo sapiens | | |
| when did equus first appear in fossil record | CGR$_{Base}$ | Evolution of the horse , Equis, Eurydice | | |
| | CGR$_{Base}$-title-only | Equidae, Equis, Euclid | | |
| | GENRE | Equidae, Equis, Equinox | | |
| rizal finished all the chapters of the novel noli me tangere in | CGR$_{Base}$ | Noli Me Tángere (novel) , Noli Me Tangere (opera), Noli Me Tangere (Bernini) | | |
| | CGR$_{Base}$-title-only | Noli me tangere, Noli Me Tángere (novel) , Noli Me Tangere (opera) | | |
| | GENRE | Noli me tangere, Non è l'inferno, Noli Me Tangere (opera) | | |

CGR$_{Base}$-title-only fails as it does not contain the information in its embeddings. See Appendix C.5 for more details about the contextualized embeddings of CGR$_{Base}$.

**Lexical overlap between query and retrieval target**    To see whether the main performance improvement of CGR over GENRE* comes from CGR leveraging the information contained in the contextualized token embedding, we check the performance of CGR$_{Base}$, CGR$_{Base}$-title-only, and GENRE* on queries that need document content to find the answer. We first run TF-IDF over all the queries of NQ dev set in KILT and divide the queries into two sets: low-overlap and high-overlap. Low-overlap is a set of queries with TF-IDF score lower than average, and high-overlap is the rest of the queries. For queries in the high-overlap set[7], all three models show high performance as it is easy to infer the correct retrieval target from the query alone even if the model does not know the document content. On the other hand, while all models show relatively lower performance for queries in the low-overlap sets[8] as the context information is required to infer the relationship between the query and the retrieval target[9], CGR$_{Base}$ shows the most strong performance. CGR$_{Base}$ shows about 7% higher performance on the low-overlap set and 5% higher performance on the high-overlap than GENRE* by leveraging the contextualized information encoded in the token embedding (Table 9 in Appendix). See Appendix C.6 to see more detailed examples of the prediction result of GENRE* and CGR$_{Base}$.

## 6    CONCLUSION

In this paper, we propose Contextualized Generative Retrieval (CGR), a generative retrieval model that utilizes contextualized embeddings (output embeddings of the language model encoder) rather than vanilla vocab embeddings while generating the target sequence. This way, the model does not rely only on the information encoded in its own model parameters but also on the information encoded in the contextualized embeddings. Experimental results show that CGR achieves significantly higher performance than vanilla generative retrieval, demonstrating the effectiveness of utilizing such non-parametric external memory during decoding. We also perform extensive ablation studies and analysis on several variants of contextualized generative retrieval models to better understand how they work.

---

[7]e.g., Q: where was the *world economic forum* held this year / Target Document: World Economic Forum

[8]e.g., Q: During which season does cape town receive rainfall / Target Document: Climate of South Africa

[9]Among the queries that all three models successfully retrieved the right retrieval target, 61% of queries are in the high-overlap set. Also, among the queries that all three models failed, 74% of queries are in the low-overlap set. Such rates show that queries in low-overlap are relatively difficult.

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

## A  MODEL DETAILS

See Figure 2a, Figure 1, Figure 2b, and Figure 2c for figures of generative retrieval model, CGR$_{Base}$, CGR$_{Async}$, and CGR$_{Contra}$, respectively.

### A.1  INFERENCE STEP

We perform a constrained beam search with prefix tree (Cao et al., 2021; Lee et al., 2022) during the inference step to assure that all generated sequences are in the corpus. The prefix tree is constructed with the tokenization result of the corpus, and we perform a constrained beam search by masking out the tokens that do not create a sub-string of the text in the corpus. We find the next tokens from the top-k of the unmasked ones. While *token ID* was used as the node of the prefix tree in previous works since each token was mapped to a unique token ID, we construct a prefix tree with the *text* of the token as the node, because CGR contains multiple token IDs for a single token. Therefore, rather than unmasking only a single token ID, we unmask all token IDs that correspond to the text in order to unmask a token. We keep the beam size to 10 for all experiments following Cao et al. (2021).

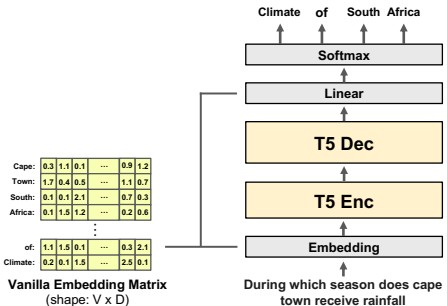

(a) Model architecture of Generative Retrieval Model. It uses the vanilla embedding matrix while generating the retrieval target.

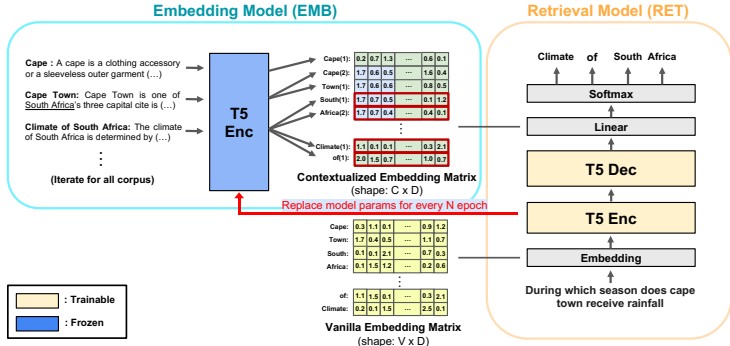

(b) Model architecture of CGR$_{\text{Async}}$. It differs from CGR$_{\text{Base}}$ in that EMB is replaced by the encoder of RET every $N$ epochs.

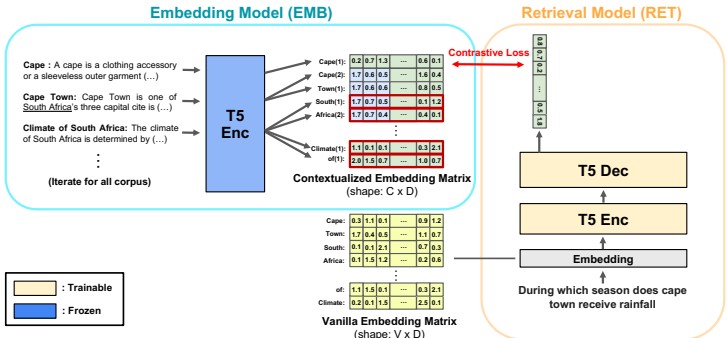

(c) Model architecture of CGR$_{\text{Contra}}$ step1 (contrastive learning). The first output token embedding from the decoder of RET is trained to be close to all contextualized token embeddings of the retrieval target (positive pairs) but far from other contextualized token embeddings (negative pairs). Note that the model architecture of CGR$_{\text{Contra}}$ step2 (generative retrieval task) is the same as CGR$_{\text{Base}}$ (Figure 1).

Table 6: R-precision(%) for the document retrieval task on NQ and TQA test dataset in KILT. See Appendix C.2 for details about how the loss term differs. The loss term is used while training $CGR_{Contra}$ in contrastive learning (step 1 of training $CGR_{Contra}$).

| Positive | Negative | NQ | TQA |
|---|---|---|---|
| Single Token Emb | In-Batch Negatives | 60.0 | 68.9 |
| Single Token Emb | Contextualized Embedding Matrix | 58.9 | 68.4 |
| Multiple Token Emb | Contextualized Embedding Matrix | 60.3 | 68.9 |

# B EXPERIMENTAL SETUP

## B.1 CGR & GENRE*

We train all models using a pre-trained T5-large checkpoint from Wolf et al. (2020) as the initial checkpoint. GENRE* and CGR are trained with the same hyperparameter setting for a fair comparison. We experiment over 8 32GB V100 GPUs or 2 48GB A6000 GPUs. We train using Adafactor with a learning rate 1e-4 with a linear warm-up for the first 10% of training and then linear decay with batch size 512 till a maximum of 150 epochs.

## B.2 BM25 & DPR

Unlike Maillard et al. (2021), which performs document retrieval tasks by training the model on passage-level tasks and considers retrieval successful when it retrieves the document that contains the passage, to match the setting (dataset) similar to CGR, we train the model in document retrieval task. We consider the first five paragraphs as the content of the document and train the model so that the query embedding gets close to not the paragraph embedding but the document embedding. The number of the corpus in the document retrieval tasks are the same as the number of page in the KILT dataset.

For BM25, the corpus is the same as in DPR where each item in the corpus is the first five paragraphs of individual documents in the KILT corpus.

# C EXPERIMENTAL RESULTS

## C.1 UPDATE FREQUENCY IN $CGR_{ASYNC}$-TITLE-ONLY

We analyzed how the performance changes according to how often the replacement of EMB by the encoder of RET occurs (replacement for every $N$ epoch) with $CGR_{Async}$-title-only. When comparing the performance with $N = \{10, 20, 50\}$, $CGR_{Async}$-title-only shows the highest performance at $N = 10$, and the performance tends to deteriorate as N becomes larger. Also, all $CGR_{Async}$-title-only show higher performance than $CGR_{Base}$-title-only (CGR-title-only without any replacement). Results show that although the model requires high computation cost and longer training time as $N$ gets smaller, it is important to have high coherency between the contextualized embeddings (output embeddings of EMB) and RET by frequent replacement.

## C.2 DIFFERENT CONTRASTIVE LOSS IN $CGR_{CONTRA}$

We experiment with three different types of contrastive loss when training $CGR_{Contra}$. In this section, we show the losses and how the results differ by such methods.

Given a training dataset of pairs $\{(\boldsymbol{q}, \boldsymbol{t})\}$ where $\boldsymbol{q}$ is the query text, and $\boldsymbol{t}$ is the retrieval target (title of the document to retrieve) composed of multiple tokens $\boldsymbol{t}_i$ ($1 \leq i \leq k$ where $k$ is the length of the target), we split all tokens into $k$ separate pairs $\{(\boldsymbol{q}, \boldsymbol{t}_i)\}$ to construct the training dataset of query-token. The three loss differs in what the model considers as a negative set and a positive set.

**Loss 1: Neg: In-Batch Negatives / Pos: Single Token Embedding**  With the query-token dataset, we train RET's first output token representation from the decoder to be close to all $\mathbf{t}^+ \in \{\mathbf{t}_1, \cdots, \mathbf{t}_k\}$ (embedding of any token in the retrieval target $\boldsymbol{t}$) given the query $\boldsymbol{q}$ as an input to RET. The objective is to minimize the contrastive loss to make the query text embedding $\mathbf{q}$ be closer to positive token embedding $\mathbf{t}^+$:

$$L(\mathbf{q}, \mathbf{t}^+, \mathbf{t}_1^-, \cdots, \mathbf{t}_{|\mathcal{T}^-|}^-) = -\log \frac{e^{<\mathbf{q}, \mathbf{t}^+>}}{e^{<\mathbf{q}, \mathbf{t}^+>} + \sum_{\mathbf{t}^- \in \mathcal{T}^-} e^{<\mathbf{q}, \mathbf{t}^->}} \tag{3}$$

where $\langle \, , \, \rangle$ is the inner product value between the two embeddings, and $\mathcal{T}^- = \{\mathbf{t}_1^-, \cdots, \mathbf{t}_{|\mathcal{T}^-|}^-\}$ is the set of negative token embeddings, which are other token embeddings in the training batch that are not paired with $\boldsymbol{q}$ (in-batch negatives (Karpukhin et al., 2020)).

**Loss 2: Neg: Contextualized Embedding Matrix / Pos: Single Token Embedding**  The loss differs from the upper loss in that it considers *all* embeddings in contextualized embedding matrix except the single positive embedding as negative rather than performing the in-batch negatives which consider the subset of contextualized embedding matrix as negatives. The equation is same as Equation 3, but elements in $\mathcal{T}^-$ are *all* other token embeddings in contextualized embedding matrix.

**Loss 3: Neg: Contextualized Embedding Matrix / Pos: Multiple Token Embedding**  The loss differs from the upper loss in that it considers *all* token embeddings in the title as positive embeddings; for each query $\boldsymbol{q}$, there are more than one positive contextualized token embeddings.

With the query-token dataset, where $\mathcal{T}^+ = \{\mathbf{t}_1^+, \cdots, \mathbf{t}_k^+\}$, set of positive token embeddings, we train RET's first output token representation from the decoder to be close to all token embeddings in $\mathcal{T}^+$ given the query $\boldsymbol{q}$ as an input to RET. The objective is to minimize the contrastive loss to make the query text embedding $\mathbf{q}$ be closer to all positive token embedding in $\mathcal{T}^+$:

$$L(\mathbf{q}, \mathbf{t}_1^+, \cdots, \mathbf{t}_{|\mathcal{T}^+|}^+, \mathbf{t}_1^-, \cdots, \mathbf{t}_{|\mathcal{T}^-|}^-) = -\log \frac{\sum_{\mathbf{t}^+ \in \mathcal{T}^+} e^{<\mathbf{q}, \mathbf{t}^+>}}{\sum_{\mathbf{t}^+ \in \mathcal{T}^+} e^{<\mathbf{q}, \mathbf{t}^+>} + \sum_{\mathbf{t}^- \in \mathcal{T}^-} e^{<\mathbf{q}, \mathbf{t}^->}} \tag{4}$$

where $\langle \, , \, \rangle$ is the inner product value between the two embeddings, and $\mathcal{T}^- = \{\mathbf{t}_1^-, \cdots, \mathbf{t}_{|\mathcal{T}^-|}^-\}$ is the set of negative token embeddings, which are *all* other token embeddings in contextualized embedding matrix.

**Results**  Table 6 show the performance of CGR$_{\text{Contra}}$ with different contrastive loss by what it considered as the positive pair and the negative pair. *Multiple Token Emb* considers all token embeddings in the same target sequence as positive pairs, and *Single Token Emb* considers all token embeddings separately thus only one of the token embedding from the title token embeddings is considered as positive pair. *In-Batch Negatives* considers all embeddings in a batch except for the positive embedding as negative pairs, and *Contextualized Embedding Matrix* considers all embeddings in the contextualized embedding matrix (a matrix constructed with the contextualized token embeddings) except for the positive embeddings as negative pairs.

The model trained on contrastive loss with multiple token embeddings as positive pairs, and all other embeddings in contextualized embedding matrix as negative pairs (Loss3) show the highest performance. The model trained on the same negative but with a single token embedding as positive (Loss2) shows the lowest performance. The model with single token embedding as positive and in-batch negatives as negative pairs (Loss1) shows the performance in-between.

As in Xiong et al. (2021), the model with Loss2 and Loss3 has the benefits of looking at the global embedding space by considering the contextualized embedding matrix as the negative pair, unlike Loss1 which only considers embeddings in the same batch as negatives (in-batch negatives). However, Loss2 show lower performance than Loss1 as in the case where the model considers a single token embedding as a positive pair, the model considers the rest of the token embeddings in the same title as the negative pair. As the token embeddings in the same title are matched with the same query, such a training method seems to make the model confused and leads to bad performance. Thus when considering a single token embedding as positive pair (Loss1 or Loss2), it is better to consider only the embeddings in the same batch as negatives (in-batch negatives) rather than on all the token embeddings (Contextualized Embedding Matrix) as there is a low possibility of the model to have two different token embeddings of the same title in a batch.

### C.3  CLUSTERING

**Example of Clustering**  When a token "the" appears in the corpus 100 times, 100 different contextualized embeddings of "the" are extracted by the encoder model at first. Then, we perform k-means clustering on the 100 contextualized embeddings to cluster them into at most $k$ clusters and save all centroid embeddings. We leave only the $k$ centroid embeddings as the decoder vocab embeddings of the token "the" and assign a new decoder token ID for each contextualized embedding by the cluster it belongs to. By repeating the process over all the tokens, each token has a number of contextualized embeddings less or equal to $k$. As there are multiple contextualized token embeddings for a single token, we replace the ground-truth target token IDs with the newly constructed decoder token IDs to specify which contextualized token embedding the ground-truth target token ID is referring to.

**Performance by Number of Clusters and Storage Footprint**  As saving all contextualized token embeddings to use as the vocab embedding matrix requires a large storage footprint ($\approx$ 148GB), we reduce the number of token embeddings by clustering and saving only the $k$ centroid embeddings for each token (Section 4.4). Figure 5 shows the effect of the maximum number of clusters for each token ($k$) on the performance. Models with a $k = 5$ (maximum of five different contextualized token embeddings for each token) show the highest performance and having $k$ smaller or larger than five decreases the performance. We hypothesize that performance of models with $k < 5$ degrades because the number of the embeddings is too small to contain all different contextual meanings of the token and thus will be closer to vanilla token embedding. In contrast, the performance of models with $k > 5$ decreases because the search space of each generation step is too large and the parametric space of the model becomes too fine-grained which might distract the model. When $k = 5$, the number of embeddings is 980 times less than using all the contextualized embeddings of KILT corpus as the vocab embeddings and 3.7 times larger than using the vanilla vocab embeddings of T5 ($k = 1$). Therefore, when $k = 5$, it needs 0.47GB of storage footprint to save all the vocab embeddings, whereas the vanilla vocab embeddings ($k = 1$) need 0.13GB. The increase in the storage footprint of vocab embeddings (0.34GB) is marginal compared to the storage footprint to save the model parameters (3GB).

### C.4  CHARACTERISTICS OF CONTEXTUALIZED EMBEDDINGS IN VARIANTS OF CGR

We compare the contextualized token embeddings of $CGR_{Base}$, $CGR_{Async}$, and $CGR_{Contra}$[10] For 1000 cluster embeddings, we check the rate of the same token among the top-5 embeddings similar to the corresponding embedding. $CGR_{Base}$ shows the lowest rate of 50%. $CGR_{Async}$ and $CGR_{Contra}$ show a similarly high rate of 70%. The rate tends to increase as $N$ increases in $CGR_{Async}$. Such results suggest that as same token has similar lexical meaning, it is better to have a relatively similar meaning. However, as the performance increases as a single token are matched to multiple token embeddings till $k = 5$ (Appendix C.3), it is is also

---

[10]We analyze the EMB of step2 in $CGR_{Contra}$ and last replace EMB for $CGR_{Async}$.

important to have slightly different meanings depending on the surrounding context. When checking which corpus bundles are bound to the same cluster, all three tend to depend on which *position* of text the token is placed on and the *meaning* of surrounding tokens. See Appendix C.5 for more details.

### C.5 CLUSTERING OVER TOTAL EMBEDDINGS

To understand the spatial properties of the contextualized embeddings, we conducted a qualitative analysis on the embeddings, by performing k-means clustering over the total contextualized token embeddings of $CGR_{Base}$ (EMB is the encoder of T5-large). Specifically, we clustered 36 million token embeddings, obtained from EMB, into 117,508 clusters[11] using the FAISS k-means library (Johnson et al., 2021).

First, we randomly sampled 100 tokens, and for each token, we calculated the portion of the contextualized embeddings that belong to the top 10% of the clusters which contain the most embeddings of the token. As a result, on average 67.6% of the embeddings of a token are contained in the 10% of the clusters which contain the token, with a standard deviation of 22.7. This indicates that most of the tokens are concentrated in a few spatial regions, while the others are spread over many different areas.

To get a deeper insight into the spatial properties of the embeddings, we picked two tokens, "Lincoln" and "Squad" and visualized some of the clusters that contain the tokens(Table 7). For each cluster, the tokens belonging to the cluster and their corresponding document names are shown. In (Table 7), at most 20 documents are shown for each token and only 4 tokens are shown in cluster 3 and 4 for simplicity. The first and second examples show the case that a cluster is composed of only a single token, as mentioned above. Interestingly, all of the corresponding documents of the first cluster are related to Lincolnshire, a county of England. Similarly, the tokens in the second cluster are related to the documents about sports (usually football) squads. On the other hand, the third and fourth examples show the other case that a cluster contains only a few tokens that we are interested in. The members of the third cluster are related to the middle names, and a few embeddings of the token "Lincoln" is contained in this cluster since there are some Wikipedia documents of the people whose middle name is Lincoln. Likewise, the fourth cluster consists of the embeddings which are related to the name of music albums(usually hip-hop and rock), where some of them are produced by the group named "Blazin' squad", for example. These examples show how expressive can the contextualized embeddings be compared to the vanilla token embeddings; in this case, it is hard to expect that these various context-dependent information of a token can be sufficiently encoded into a single token embedding.

In summary, the results show that the contextualized embeddings corresponding to the same token are mapped to many different regions of the embedding space, depending on its context. This implies that the contextualized embeddings successfully acquired the contextual information of the corresponding documents, highlighting the effectiveness of utilizing contextualized embeddings for generative retrieval.

### C.6 LEXICAL OVERLAP BETWEEN QUERY AND ANSWER

CGR show especially strong performance on queries in the low-overlap set; queries that in most cases need the context information unless the model saw the information during the training step (Section 5.2). We check four sets:
1. GENRE+/CGR +: queries where both CGR and GENRE* successfully retrieved
2. GENRE+/CGR-: queries where GENRE* successfully retrieved and CGR failed
3. GENRE-/CGR +: queries where GENRE* failed and CGR succeed
4. GENRE-/CGR-: queries where GENRE* and CGR both failed.

---

[11]The number of the clusters is same as the number of the tokens in contextualized embedding matrix, hence same as the number of the clusters we used in 4.4.

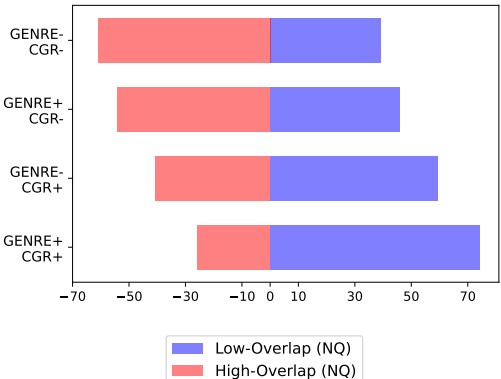

Figure 3: Red bar indicates the high-rate and the blue bar indicates the low-rate. The rate is measured by NQ dev set in KILT. Details about high-rate and low-rate is in Appendix C.6.

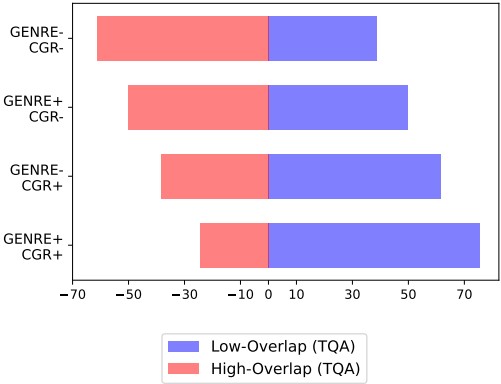

Figure 4: Red bar indicates the high-rate and the blue bar indicates the low-rate. The rate is measured by TQA dev set in KILT. Details about high-rate and low-rate is in Appendix C.6.

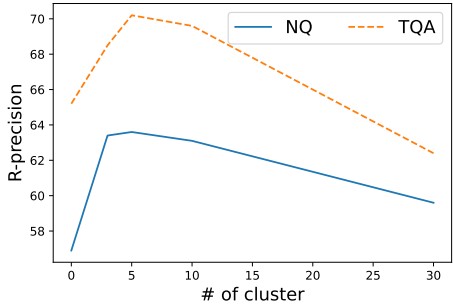

Figure 5: Effect of number of cluster in Base CGR. Results of zero number of clusters are from GENRE*.

Table 7: Examples of clusters when clustering over total contextualized token embeddings when EMB is the encoder of T5-large.

| Cluster | Token | Documents |
|---|---|---|
| 1 | _Lincoln | Moulton, Lincolnshire / Belton, North Lincolnshire / Walcott, Lincolnshire / Wrangle, Lincolnshire / Swineshead, Lincolnshire / Leverton, Lincolnshire / Kirton, Lincolnshire / Benington, Lincolnshire / Bicker, Lincolnshire / Dyke, Lincolnshire / Hilldyke, Lincolnshire / Waltham, Lincolnshire / Reepham, Lincolnshire / Bradley, Lincolnshire / Allington, Lincolnshire / Donington, Lincolnshire ettleton, Lincolnshire / Panton, Lincolnshire / Beckingham, Lincolnshire / Bigby, Lincolnshire / ... |
| 2 | _Squad | Field hockey at the 2000 Summer Olympics – Men's team squads / Field hockey at the 2004 Summer Olympics – Men's team squads / Field hockey at the 1996 Summer Olympics – Men's team squads / Football at the 2000 Summer Olympics – Men's team squads / Football at the 1996 Summer Olympics – Men's team squads / List of Queensland rugby league team squads / Football at the 2006 Lusophony Games – Men's team squads / List of current AFL team squads / Football at the 1912 Summer Olympics – Men's team squads / List of New South Wales rugby league team squads / Football at the 1996 Summer Olympics – Women's team squads / Football at the 1988 Summer Olympics – Men's team squads / Football at the 1984 Summer Olympics – Men's team squads / Football at the 1976 Summer Olympics – Men's team squads / Football at the 1900 Summer Olympics – Men's team squads / Football at the 1904 Summer Olympics – Men's team squads / Football at the 1908 Summer Olympics – Men's team squads / Football at the 1992 Summer Olympics – Men's team squads / Football at the 1980 Summer Olympics – Men's team squads / Football at the 1972 Summer Olympics – Men's team squads / ... |
| 3 | _Lincoln | William Lincoln Garver / Albert Lincoln Washburn / Charles Lincoln Edwards / Thomas Lincoln Casey Sr. / James Lincoln Collier / Abraham Lincoln Lewis / Earl Lincoln Poole / George Lincoln Goodale / George Lincoln Burr / Abraham Lincoln Keister / Elmer Lincoln Irey / Walter Lincoln Hawkins / Abram Lincoln Harris / Abraham Lincoln DeMond / Thomas Lincoln Tally / Abraham Lincoln Filene / Mary Lincoln Beckwith / Frederick Lincoln Emory / Howard Lincoln Hodgkins / Oliver Lincoln Lundquist / ... |
| | _Levi | John Levi Marti / John Levi Sheppard / Moses Levi Ehrenreich / Nathaniel Levi Gaines / Harry Levi Hollingworth / Thomas Levi Whittle / Austin Levi Fraser / George Levi Crane / Olin Levi Warner |
| | _Luke | Milledge Luke Bonham / Henry Luke Orombi / Henry Luke White / Vincent Luke Palmisano / George Luke Smith / Henry Luke Bolley / Mary Luke Tobin / James Luke Prendergast / John Luke Lowther / Jerry Luke LeBlanc / Thomas Luke Msusa / Robert Luke Deakin / Joseph Luke Cecchini |
| | _Lane | Carroll Lane Fenton |
| | ... | |
| 4 | _Squad | True Story (Terror Squad album) / The Album (Terror Squad album) |
| | _Angel | Covenant (Morbid Angel album) / Domination (Morbid Angel album) / The Art of Dying (Death Angel album) / Act III (Death Angel album) / Heretic (Morbid Angel album) |
| | _Butterfly | Heavy (Iron Butterfly album) / Metamorphosis (Iron Butterfly album) / Ball (Iron Butterfly album) |
| | _Flip | Flip-flop (electronics) / Flipper (anatomy) / Respect Me (Lil' Flip album) / The Leprechaun (Lil' Flip album) |
| | ... | |

Note that CGR is CGR$_{Base}$ in this section. Figure 3 and Figure 4 show the low-rate (blue) and high-rate (red). Low-rate of each case is calculated as $\frac{\{Q \cap L\}}{Q}$, where $Q$ is a set of queries in each case and $L$ is a set of queries in a low-overlap set. High-rate of each case is calculated as $\frac{\{Q \cap H\}}{Q}$, where $H$ is a set of queries in a high-overlap set.

Table 8: Top-3 prediction result of CGR$_{Base}$, and GENRE*

| Query | Prediction Result |
|---|---|
| what do the 3 dots mean in math | **CGR$_{Base}$**    Therefore sign , Infinity symbol, Equation |
| | **GENRE**    Ellipsis, Infinity symbol, Homo sapiens |
| what does the pearl symbolize in the bible | **CGR$_{Base}$**    Parable of the Pearl , Mitzvah, Pearl of Wisdom |
| | **GENRE**    Pearl of Great Price, Perlin, Promised Land |
| does archie end up with betty or veronica in riverdale | **CGR$_{Base}$**    Archie Marries Veronica/Archies Marries Betty , List of Riverdale characters, Archie Buchanan |
| | **GENRE**    Riverdale (2017 TV series), List of Riverdale characters, Archie Mitchell |
| actor who plays dr avery on grey's anatomy | **CGR$_{Base}$**    Jesse Williams (actor) , Jesse Williams, Jesse Spencer |
| | **GENRE**    Marc Alaimo, Patrick Warburton, Jeffrey Dean Morgan |
| when did equus first appear in fossil record | **CGR$_{Base}$**    Evolution of the horse , Equis, Eurydice |
| | **GENRE**    Equidae, Equis, Equinox |
| who decides the number of judges in the high court | **CGR$_{Base}$**    Indian High Courts Act 1861 , High Court of Australia, Supreme Court of India |
| | **GENRE**    Supreme Court of the United Kingdom, Supreme Court of India, High Court of Australia |
| when's the last time the philadelphia eagles played the new england patriots | **CGR$_{Base}$**    Super Bowl XXXIX , New England Patriots, Super Bowl XXXVIII |
| | **GENRE**    New England Patriots, Philadelphia Eagles, History of the Philadelphia Eagles |
| rizal finished all the chapters of the novel noli me tangere in | **CGR$_{Base}$**    Noli Me Tángere (novel) , Noli Me Tangere (opera), Noli Me Tangere (Bernini) |
| | **GENRE**    Noli me tangere, Non è l'inferno, Noli Me Tangere (opera) |
| during which season does cape town receive rainfall | **CGR$_{Base}$**    Climate of South Africa , City of Cape Town, Cape Town water crisis |
| | **GENRE**    Cape Town, City of Cape Town, Cape Town water crisis |

Table 9: R-precision(%) for the document retrieval task on NQ dev dataset in KILT. See details about Low and High Overlap in Section 5.2.

| | GENRE* | CGR$_{Base}$-title-only | CGR$_{Base}$ |
|---|---|---|---|
| Low-Overlap | 45.8 | 51.6 | 52.7 |
| High-Overlap | 71.3 | 75.3 | 75.8 |
| Total | 58.3 | 63.2 | 64.0 |

For both figures, GENRE-/CGR + shows a higher number in low-rate, which indicates that CGR tend to successfully predict queries in the low-overlap set compared to GENRE*. Also, for both figures, GENRE+/CGR + shows a high number in low-rate and GENRE-/ours- shows a high number of high-rate, which indicates that queries in the high-overlap set tend to be easy questions for both GENRE and CGR whereas queries in the low-overlap set are difficult for both models.

Also, Table 8 shows samples of the top-5 prediction results of CGR$_{Base}$ and GENRE* where CGR$_{Base}$ successfully retrieved the correct item and GENRE* failed. Such queries tend to be in the low-overlap set. Such results suggest that CGR (CGR$_{Base}$) is robust on queries in the low-overlap set compared to GENRE*.

| | No Training | | Single ($\leq 3\%$) | | | | All KILT (100%) | | | |
|---|---|---|---|---|---|---|---|---|---|---|
| Dataset | BM25 | DPR | GENRE* | $CGR_{Base}$ | $CGR_{Async}$ | $CGR_{Contra}$ | DPR | GENRE | $SEAL_s$ | $SEAL_i$ |
| NQ | 25.8 | 63.2 | 59.9 | 63.4 | 63.7 | **63.9** | 62.3 | **64.3** | 64.2 | |
| TQA | 29.4 | 65.1 | 64.2 | 67.5 | 67.9 | **68.7** | 62.0 | **71.1** | 68.3 | 68.3 |

Table 10: R-precision(%) for document retrieval task on NQ and TQA dev dataset in KILT. Results of BM25 and DPR are from Maillard et al. (2021), results of SEAL are provided by the authors of Bevilacqua et al. (2022), and results of GENRE are from the released pre-trained models. $SEAL_s$ is SEAL (LM+FM) and $SEAL_i$ is SEAL (LM+FM, intersective) in Bevilacqua et al. (2022). Underlined model is the direct baseline of CGR. Best of each section (same training dataset) in **bold.**

Table 11: Hits@1 and Hits@10 in NQ-320k. Results of BM25, DSI-Naive, and DSI-Semantic are from Tay et al. (2022). $CGR_{Base}$-Naive is $CGR_{Base}$ with document ID as a retrieval target, and $CGR_{Base}$-Title is $CGR_{Base}$ with a title of the document as a retrieval target. Underlined model is direct baseline of $CGR_{Base}$-Naive. Best from document ID as a target sequence in **bold**.

| | BM25 | DSI-Naive | DSI-Semantic | $CGR_{Base}$-Naive | $CGR_{Base}$-Title |
|---|---|---|---|---|---|
| Hits@1 | 11.6 | 13.3 | 35.6 | **58.7** | 62.2 |
| Hits@10 | 34.4 | 33.6 | 62.6 | **73.1** | 78.8 |

## C.7 KILT Dev Results

Table 10 shows the result of five models (BM25, GENRE, DPR, SEAL, and CGR) in the NQ and TQA of the KILT dev set. Note that the DPR and BM25 models in the single setup[12] are different from Table 1 (Section B.2). As in results with KILT test datasets (Table 1), results with KILT dev datasets (Table 10) show similar trends. CGR shows an average of 7% higher performance compared to its direct baseline model, GENRE*. $CGR_{Contra}$ shows the highest performance over three variants of CGR. When comparing the results with other retrieval models, CGR shows the highest performance in the single setup and shows comparable performance to models trained with all KILT datasets although CGR is trained with only 3% of the training dataset.

## C.8 Contextualized Embeddings with document ID as retrieval target

To show that CGR is not restricted to the title but is generalizable to various retrieval targets, we experiment CGR ($CGR_{Base}$-Naive) when the retrieval target is random document ID (Naively Structured String Identifiers in DSI (Tay et al., 2022)) in NQ-320k. In the case, direct baseline model to $CGR_{Base}$-Naive is DSI-Naive (DSI with Naively Structured String Indentifiers as the document ID) not GENRE*[13]. From Table 11, we can see that $CGR_{Base}$-Naive shows more than four times higher performance in Hits@1 compared to DSI-Naive.

Also, $CGR_{Base}$-Naive shows higher performance than DSI-Semantic where the document ID of DSI-Semantic is built by the hierarchical semantic meaning inside the document content. The results show the importance of using the contextualized embeddings as the decoder vocab embeddings; rather than building the document ID using the document content, using a random document ID but with the document content inside the token embedding of each document ID shows higher performance.

---

[12]Single setup is a setup where the models are trained with a single dataset; NQ only or TQA only.

[13]Note that the document ID of $CGR_{Base}$-Naive and DSI-Naive are not exactly the same as the document ID of DSI is not released. However, both the document ID of $CGR_{Base}$-Naive and DSI-Naive are the same in that they are constructed randomly. We plan to update the numbers when the official repo is open

Moreover, from the results between CGR$_{Base}$-Naive and CGR$_{Base}$-Title, we can see that the retrieval target affects the performance. We assume the difference comes from whether the retrieval target is a natural language text or not; CGR can leverage the benefits of pre-training step more when the target sequence is natural language (CGR$_{Base}$-Title). However, the difference between CGR$_{Base}$-Naive and CGR$_{Base}$-Title is marginal compared to the performance difference between DSI and GENERE where the two models both used vanilla model embeddings, which suggests that CGR is generalizable to various retrieval targets. We leave other target sequences apart from the title and the document ID as future work.

When compared to the performance of DSI T5-XXL with Semantic String Docid (the best performing model in Tay et al. (2022)) (40.4), CGR$_{Base}$ (58.7) shows about $1.5\times$ higher performance in NQ-320k Hits@1. Note that the model has 14 times more parameters than CGR$_{Base}$.

## D  LIMITATIONS & FUTURE WORKS

As CGR uses k-means clustering to reduce the number of contextualized embeddings, the performance may change by how the contextualized embeddings are clustered. Generative retrieval models show low performance to unseen cases due to their dependency on the parametric space of the model (Lee et al., 2022); the model is likely to retrieve sequences it has seen during the training step better as the information is saved in the parameters. Therefore, as CGR also leverages the benefit of the parametric space, we leave the direction of adapting the model to new corpus as future work.

