# OpenReview forum: "Contextualized Generative Retrieval"
_ICLR.cc/2023/Conference — Submitted to ICLR 2023_

### Official Review · Reviewer_rcjJ · 2022-10-22

**Confidence:** 4
**Correctness:** 3
**Technical Novelty And Significance:** 2
**Empirical Novelty And Significance:** 2
**Recommendation:** 3

**Clarity, Quality, Novelty And Reproducibility:**

The paper is unclear on the model definition, making it hard to judge the quality. Based on my inference, the approach is reasonable but not terribly surprising.

**Strength And Weaknesses:**

Strengths:
- It is a reasonable approach to injecting external knowledge in generative retrievers.

Weaknesses:
- Unclear writing: Figure 1 and a verbal description in Section 3 is all we have to decipher what the paper does. I can guess what it does, but it's never fully clear.
- The setup is questionable. GENRE uses BART-large whereas this paper uses T5, so the focus on comparing directly with GENRE seems a bit misguided.
- Limited experiments (not training on all of KILT).

**Summary Of The Paper:**

The paper proposes to pre-compute contextual token embeddings and use them in the decoder of generative retrievers (as a way to enable them to use external knowledge). The model performs favorably on the KILT retrieval tasks when trained on a subset of KILT.

**Summary Of The Review:**

The paper proposes a reasonable way to use external knowledge in generative retrievers, but the quality of innovation, experiments, and writing seems to fall below the standard of ICLR.

---

> ### Author Response · Authors · 2022-11-15
> **Response to Reviewer rcjJ**
>
> Hello Reviewer rcjJ,
>
> Thank you for your valuable time in reviewing our paper.
>
> > Unclear writing: Figure 1 and a verbal description in Section 3 is all we have to decipher what the paper does. I can guess what it does, but it's never fully clear.
>
> Thank you for your comment. We have added some extra content to the updated paper to make it more clear. We have also added some overall big points of the paper below.
>
> * The objective of the work is to bring both the benefits of bi-encoder and generative retrieval models; CGR overcomes the embedding bottleneck problem of bi-encoder retrieval by utilizing the parametric space of the model and overcomes the full dependency on parametric space as in generative retrieval by leveraging the information embedded in the contextualized embeddings.
>
> * Section3 (Contextualized Generative Retrieval) shows our big picture of how to use such contextualized generative retrieval models. In this section, we did not constrain the explanation of CGR to only the document retrieval tasks with the title as the retrieval target but show how the method can generalize to various retrieval tasks with various retrieval targets. Here are some steps about how CGR works:
>
>    * The model first encodes all contextualized embeddings of target sequences which are the output embeddings of EMB (the embedding model). Here, the target sequence could be any arbitrary sequence. In our paper, we constrain the case to the title of the document to perform a document retrieval task and compare it in the same setting with our baseline GENRE (GENRE used title as the retrieval target).
>
>     * We perform k-means clustering over all contextualized embeddings for each token to reduce the number of contextualized embeddings per token to be smaller than k; in the paper k is 5. Detailed examples of the clustering are in Appendix C.3.
>
>     * The centroid embeddings then become the contextualized embeddings when training RET (the retrieval model); when given a query, RET is trained to generate the target sequence. The embeddings of the target sequence of CGR are not vanilla model vocab embeddings but the centroid embeddings from step2.
>
> * Section4 (Model Details) shows three different architectures of CGR (CGR_base, CGR_async, CGR_contra). Section4 is detailed on document retrieval tasks, unlike Section3 which explains CGR used on general cases not limited to the document retrieval tasks. As it is the first work using only the contextualized embeddings as the model vocab during the decoding step, we first propose the most basic model, CGR_base, which extracts the contextualized embeddings with the frozen pre-trained language model. To improve the performance and to fully leverage the advantages of using contextualized embeddings, we propose two variants from the CGR_base which are CGR_async and CGR_contra. CGR_async is CGR_base with an asynchronous update on the contextualized embeddings. CGR_contra is CGR_base where the contextualized embeddings are initialized by contrastive learning. Figures about each model architecture are in Appendix A (page 12).
>
>
> * Section5 (Experiments) shows the overall results of CGR and its baseline GENRE (the generative retrieval model which uses the model vocab embeddings when decoding) in KILT (NQ, TQA) and NQ-320k. From the results, we show that using contextualized embeddings during the decoding step is a simple and effective method. We further compare the results in the zero-shot setting in Table 4, which shows that CGR tends to generalize better as it utilizes not only the information embedded (injected) into the model parameters as in previous generative retrieval models but also the non-parametric information in the contextualized embeddings. Moreover, in Table3 and Table5, to show that the non-parametric space of contextualized embeddings is important in the performance, we compared the three models (CGR_base, CGR_base-title-only, and GENRE*) where CGR_base consistently shows higher performance than CGR_base-title-only (CGR model which has limited information of title only without the document content in contextualized embeddings) and GENRE* (generative retrieval model without contextualized embeddings).

---

> > ### Author Response · Authors · 2022-11-15
> > **Response to Reviewer rcjJ**
> >
> > > The setup is questionable. GENRE uses BART-large whereas this paper uses T5, so the focus on comparing directly with GENRE seems a bit misguided.
> >
> > GENRE* is a model trained with the same setting as CGR (footnote 4), i.e. the same base pre-trained model (T5-large) and the same training recipe (dataset and hyperparameter). We have updated the paper to make the writing more clear. Please note that the results of GENRE* in Tables 3, 4, and 5 are also in the same setting as CGR.
> >
> > For Table 2, we have replaced the performance of GENRE (Bart-large) with GENRE* (T5-large) and have added the performance of BM25 and CGR_contra. Also, as mentioned in the “Common response to reviewers” comment, the performance of SEAL from the official paper seems to be underestimated, and the authors are working on resolving the issue (https://github.com/facebookresearch/SEAL/issues/14). Therefore, we have removed the performance.
> >
> > > Limited experiments (not training on all of KILT).
> >
> > We agree that it would be interesting to see the performance of CGR with the full KILT dataset. However, as mentioned in the paper, GENRE [1] uses 128 V100 GPUs for 33 hours which is not easily affordable for most institutions and companies. We performed all experiments with 8 V100 GPUs or equivalents due to limited resources. Also, as it is the first work of training with contextualized embedding in an encoder-decoder generative retrieval model, we focused on extensive ablation studies to share a comprehensive understanding of such decoding methods via contextualized embeddings.
> >
> > We would like to emphasize that even with a small number of datasets, it shows similar performance to models trained on full KILT datasets. Also, we showed that CGR tends to improve performance when the number of datasets increases in Table 1.

---

### Official Review · Reviewer_UYMW · 2022-10-24

**Confidence:** 4
**Correctness:** 4
**Technical Novelty And Significance:** 3
**Empirical Novelty And Significance:** 2
**Recommendation:** 5

**Clarity, Quality, Novelty And Reproducibility:**

The paper is clearly written. The proposed method makes sense but the contribution of adding a contextualized embedding space might be considered as light.

**Strength And Weaknesses:**

Strengths
-	Important timely topic in information retrieval.


Weaknesses
- Evaluation on datasets that are characterized by title, document ID or datasets for language tasks, and not on large-scale document retrieval collections where the specific content of the documents is important to satisfy the query, and where expressiveness and discriminative power of the document representations play an important role.
- Limited technological contribution.




**Summary Of The Paper:**

The paper regards generative retrieval that given a query generates important information about text documents from the parameter space of a trained model instead of relying on fixed-sized document embeddings which might be characterized by limited expressiveness. Such a generative model interacts with the parameters of the trained model.  The authors propose the use of a contextualized embedding space. The contextualized embedding space is created using a language model.


**Summary Of The Review:**

Information retrieval is characterized by dynamically changing document collections and their content (e.g., novel named entities, events at new locations, etc,). It is not clear how the model guarantees a correct generation of key descriptors of a document, if the generative retrieval model and the contextual language model were not trained with it. This relates to the core claim of the paper. Please clarify.

The initial representations used for training the retrieval model and the representations obtained by the language model and their postprocessing (e.g., clustering of embeddings) play a role with respect to the content that is finally retained from each document. What if these representations were changed by using alternative technologies to build them? The representations are still built with neural nets that are currently only able to precisely capture a limited context and not the content of a large document.

Could the proposed generative model be combined with older generative retrieval models that operate on bag-of-words or n-gram representations (e.g., Robertson-Sparck-Jones model, language retrieval model) in order to deal with specific content that should be generated? Overall references to older generative retrieval models could be added in related research.

The paper could be strengthened by analyzing the limitations of the work.

---

> ### Author Response · Authors · 2022-11-15
> **Response to Reviewer UYMW**
>
> Hello Reviewer UYMW,
>
> Thank you for your insightful comments.
>
> ### Weakness
> > Evaluation on datasets that are characterized by title, document ID or datasets for language tasks, and not on large-scale document retrieval collections where the specific content of the documents is important to satisfy the query, and where expressiveness and discriminative power of the document representations play an important role.
>
> We have further experimented CGR in NQ-320k using document ID as a retrieval target and have added the result in Appendix C.7 of the updated paper. Although the document ID is a randomly selected ID without any representative information and is thus generalizable to any case, CGR shows higher performance than the model trained with model vocab embeddings. Please refer to the above “Common response to reviewers” comment for more details. Also, please note that we experiment over the full corpus of the KILT dataset which has about 5M items that can be considered as large-scale document retrieval collections. Please let us know if we misunderstood your comment.
>
> > Limited technological contribution.
>
> The authors would like to highlight that this is the first work to entirely replace the decoder vocab space to contextualized embedding space from the training step thereby leveraging both the benefits of the bi-encoder approach and the generative approach. Please refer to the above “Common response to reviewers” comment for more details.
>
>
> ### Question
> > Information retrieval is characterized by dynamically changing document collections and their content (e.g., novel named entities, events at new locations, etc,). It is not clear how the model guarantees a correct generation of key descriptors of a document, if the generative retrieval model and the contextual language model were not trained with it. This relates to the core claim of the paper. Please clarify.
>
> * How does the model guarantees a correct generation of key descriptors of a document
>
> We guarantee the correct generation by using the prefix tree proposed in GENRE [1]; we restrict the model to generate the sequence only from the corpus by masking out the tokens that generate a sequence that is not in the corpus (Appendix A.1). However, prefix tree in CGR differs from prefix tree in [1] in that CGR uses the text of the token as the node of the tree not the token ID as in [1].
>
> Also, the generative retrieval model (RET) is trained with contextualized vocab embeddings extracted from the encoder model (EMB). We first construct the contextualized vocab embeddings with EMB and match the target sequence with its corresponding contextualized embedding. The constructed contextualized vocab embeddings replace the decoder vocab embeddings while training the generative retrieval model (RET). Please refer to Section 3 for more details. Moreover, EMB of CGR_async is asynchronously updated with the encoder of RET during the training step, and EMB of CGR_contra is trained with contrastive loss to construct better contextualized embedding space. Please refer to Section 4 and Appendix C.2 for more details.
>
> * How does the model generate when a new corpus is added (dynamically changing document collections)?
>
> A simple method would be to add the new corpus to the prefix tree, which the model would be able to generate the new corpus. However, as the corpus is not trained with the new corpus, it is considered unseen, and generative retrieval models showed low performance when retrieving the unseen corpus [2]. We have added the limitation in Appendix D. We leave various methods to adapt generative retrieval models to the new corpus as future work.
>
> > The initial representations used for training the retrieval model and the representations obtained by the language model and their postprocessing (e.g., clustering of embeddings) play a role with respect to the content that is finally retained from each document. What if these representations were changed by using alternative technologies to build them? The representations are still built with neural nets that are currently only able to precisely capture a limited context and not the content of a large document.
>
> We agree that other representations apart from output embeddings of language model encoder could be used to construct the contextualized vocab embeddings and is an interesting direction. Thank you for the suggestion!
>
> > Could the proposed generative model be combined with older generative retrieval models that operate on bag-of-words or n-gram representations (e.g., Robertson-Sparck-Jones model, language retrieval model) in order to deal with specific content that should be generated? Overall references to older generative retrieval models could be added in related research.
>
> It seems the question is referring to sparse retrieval models, which are not generative models as far as we understand. Would you be able to clarify your question? Thanks!

---

> > ### Author Response · Authors · 2022-11-15
> > **Response to Reviewer UYMW**
> >
> > > The paper could be strengthened by analyzing the limitations of the work.
> >
> > Thank you for the suggestion. We have added the limitation of CGR in Appendix D.
> >
> > ### References
> >
> > [1] Autoregressive Entity Retrieval (https://arxiv.org/abs/2010.00904)
> >
> > [2] Generative Multi-hop Retrieval (https://arxiv.org/abs/2204.13596)

---

> > ### Comment · Reviewer_UYMW · 2022-11-16
> > **LM retrieval models**
> >
> > Probabilistic relevance models are considered generative models.
> > See:
> >
> > Lafferty, J. & Zhai C. X. (2003). Probabilistic relevance models based on document and query generation. In  W.B. Croft & J. Lafferty (Eds.), Language Modeling for Information Retrieval (pp. 1-10). Boston: Kluwer Academic Publishers.

---

> > > ### Author Response · Authors · 2022-11-18
> > > **Response to LM retrieval models of UYMW**
> > >
> > > Hello Reviewer UYMW,
> > >
> > > Thank you for the clarification.
> > > It looks like we misunderstood your question; we thought you were referring to sparse retrieval models such as TF-IDF. We will make sure to add these early works in Related Work. Combining the n-gram-based language models and deep neural network-based language models seems to be an interesting future work, though it is worth noting that modern state-of-the-art language models are mostly purely deep neural networks, so we might not benefit much from using such sparse generative models.
> > > We would also greatly appreciate it if the reviewer can let us know whether other concerns of the reviewer have been addressed. Thanks!

---

### Official Review · Reviewer_ZLXX · 2022-10-24

**Confidence:** 4
**Clarity, Quality, Novelty And Reproducibility:** The paper is well-written.
**Correctness:** 4
**Technical Novelty And Significance:** 3
**Empirical Novelty And Significance:** 3
**Recommendation:** 6

**Strength And Weaknesses:**

Strength:

1. The proposed architecture is simple and elegant.

2. The proposed approach shows promising results on downstream tasks.


Weaknesses:

1. The proposed model follows the framework of generative retrieval models but applies contextualized token embeddings.  Such contextualized embedding methods have been applied in KNN-LMs. The novelty contribution is somehow limited.

**Summary Of The Paper:**

This paper proposes a contextualized generative retrieval model.  The generative retrieval model usually performs worse on unseen data and the traditional KNN retrieval model has the advantage of good generalization but also fails on documents with long sequences. The authors propose a new architecture to overcome these problems.  The proposed model uses the output of a language model encoder as vocab embeddings. The retrieval model then retrieves a target sequence by using the vocab embeddings as the decoder embeddings. Experiments show that the proposed model achieves promising retrieval performance.

**Summary Of The Review:**

This paper proposes a new generative retrieval model and the proposed model shows promising results.

---

> ### Author Response · Authors · 2022-11-15
> **Response to Reviewer ZLXX**
>
> Hello Reviewer ZLXX,
>
> Thank you for your valuable time in reviewing our paper.
>
> > The proposed model follows the framework of generative retrieval models but applies contextualized token embeddings. Such contextualized embedding methods have been applied in KNN-LMs. The novelty contribution is somehow limited.
>
> Thank you for the comment. We agree that CGR and kNN-LMs are similar in that both utilize contextualized embeddings and improve performance.
>
> However, as mentioned in the Related Works section of the paper and the “Common response to reviewers” comment, there are some major differences between the two models. Below, we have added some more details.
>
> 1. The objective of using contextualized embeddings in kNN-LM and CGR is different. The objective of kNN-LM is to reduce the perplexity without additional training datasets and shows that the model is especially strong in predicting rare patterns such as factual knowledge. As it tries to use the contextualized embeddings to find the token with a similar context in the front (content till the previous tokens), it extracts the contextualized embeddings from the trained decoder-only model which is a uni-directional context during only the inference step. However, as the objective of CGR is to encode the whole content information (e.g., the content of the document, paragraph) to the short representative words (e.g., title, document ID) to improve the retrieval performance while generating only the short representative words by utilizing the contextual information encoded in the contextualized embeddings, we use the encoder model (EMB), a separate model from the retrieval model (RET), to encode the nearby context (bidirectional information of not only the previous token but also the next tokens). The extracted contextualized embeddings are used in both the training and the inference step.
>
> 2. KNN-LM has one module (the decoder-only language model) as it uses the contextualized embeddings only during the inference time. However, CGR has two submodules (the encoder model that extracts contextualized embeddings (EMB) and the generative retrieval model (RET)) as it uses the contextualized embeddings in both the training and the inference time; EMB to extract the contextualized embeddings which is used during the training step of RET. Using two submodules is necessary to train RET with the contextualized embeddings (extracted from EMB). Therefore, we searched for how to design each submodule and how to connect the two to increase the overall performance. We explored three different CGR (CGR_base, CGR_async, CGR_contra) which differ by how the two submodules are connected to each other and how the contextualized embeddings are extracted. From various results, we could see that using the contrastive learning method to train EMB and initialize RET (CGR_contra) shows the highest performance.
>
> 3. To the best of our knowledge, CGR is the first work to entirely replace the decoder vocab space from model vocab embeddings with contextualized embedding space from the training step.

---

### Official Review · Reviewer_DJS8 · 2022-10-25

**Confidence:** 4
**Correctness:** 3
**Technical Novelty And Significance:** 3
**Empirical Novelty And Significance:** 3
**Recommendation:** 6

**Clarity, Quality, Novelty And Reproducibility:**

The proposed method is novel and intuitive. The authors could improve the clarity — e.g., the authors can make it clearer what text they put into the encoders for the contextual embeddings (in the paper they only vaguely mention that they are titles). Figure 1 is a bit confusing to me (especially Cape(1), Cape(2)). The experiment setup can be significantly improved. From the current experiments, it is hard to draw a clear conclusion.


**Strength And Weaknesses:**

# Strength

The proposed idea is novel: it combines the advantage of both autoregressive generation (GENRE, DSI, SEAL) and contextual embeddings of databases (DPR). The use of token contextual embeddings is also smart since it can be compressed by the clustering algorithm, without too much loss in information. The study of different variants of training is comprehensive. Experiment results (if trustworthy) are strong.

# Weakness

The main weakness comes from the experiment setup. The main KILT results only have the original GENRE (using a different pre-trained model, BART) and CGR (even though the authors listed DPR and SEAL, the numbers are not in the same comparable settings). The authors should provide a reproduced GENRE with the same pre-trained model and the same training recipe, as well as other baselines like BM25, a dense retrieval method (like DPR), and, if possible, SEAL.

I have some other concerns in Table 2 and Table 3. In Table 2, GENRE, DSI, and SEAL’s performance is so low on NQ-320K. I would appreciate it if the authors could clarify the setting (sorry if I missed it) or maybe retrain the GENRE on NQ-320K in the same setting as CGR (it shouldn’t be difficult since the code should be similar to CGR).

**Summary Of The Paper:**

The proposed model targets a Wikipedia page-level retrieval task: given a query, return the page (or in their case, the Wikipedia title following GENRE). Like GENRE, the authors use a sequence-to-sequence model, take the query text as input, and directly generate the title in an autoregressive manner. The innovation is instead of using static token embeddings in autoregressive decoding, the proposed model uses contextual embeddings, which are acquired by encoding all tokens in possible titles (with the Wikipedia content as context). The embeddings naturally contain both the token information like the static word embeddings and the passage information through the encoder. Arguably, the new model has both the power of autoregressive generation and the external knowledge of Wikipedia through contextual embeddings.

The authors explored several variants. First, to compress the huge number of contextual embeddings, they conduct clustering for each token’s corresponding embeddings and keep k=5 clusters. The authors also have different versions where the contextual embeddings are fixed at the beginning (CGR-base), the embeddings are updated by the updated seq2seq model (CGR-async), and the embeddings are jointly trained through contrastive learning (CGR-contra), though the differences are small.

The authors mainly conduct experiments on KILT and NQ320K, and mainly compare their model to the direct baseline, GENRE. The results show that CGR (the proposed model) significantly outperformed GENRE on KILT (especially in-domain datasets NQ and TriviaQA), and outperformed a series of other autoregressive retrieval models on NQ320K. However, one questionable point: GENRE uses BART and CGR uses T5, but the authors only compared to the original GENRE numbers in KILT experiments instead of reproducing GENRE with the same setup, which might lead to unfair comparison. There should also be other reproduced baselines like DPR (or any dense retrieval methods) and SEAL in KILT for a more comprehensive understanding.


**Summary Of The Review:**

The proposed method -- using contextual embeddings instead of static token embeddings in decoder for autoregressive retrieval -- is novel, intuitive, and inspiring. However, the experiment has significant flaws (for the most direct and important baseline, the authors took numbers from the original implementation that uses even a different pre-trained model, which leads to unfair comparison). I am leaning toward rejection. However, if the authors can update the results with the reproduced GENRE baseline, I am willing to raise my score.

---

> ### Author Response · Authors · 2022-11-15
> **Response to Reviewer DJS8**
>
> Hello Reviewer DJS8,
>
> Thank you for your valuable time in reviewing our paper. We appreciate your detailed comments about our paper.
>
> ### Weakness
> > The authors should provide a reproduced GENRE with the same pre-trained model and the same training recipe, as well as other baselines like BM25, a dense retrieval method (like DPR), and, if possible, SEAL.
>
> GENRE* is a model trained with the same setting as CGR (footnote 4), i.e. the same base pre-trained model (T5-large) and the same training setting. We have updated the paper to make the writing more clear. Please note that the results of GENRE* in Tables 3, 4, and 5 are also in the same setting as CGR.
>
> Also, we have added more baselines (BM25 and DPR) to Table 1 for easier comparison. We have kept the setting as similar as possible. Details of the settings are in Appendix B.2 of the revised paper. We have tried to reimplement SEAL with T5-large to add as a baseline, but although the code is public, it has been nontrivial to convert the code to use T5-large, and it seems the experiments are not likely to finish before the rebuttal period ends. However, please note that we have asked the author of SEAL and he has replied that SEAL showed higher performance when trained with BART-large than with T5-large.
>
> Moreover, please note that our intention of adding various baselines in Table 1 and Table 2 apart from the direct baseline GENRE* was to give a more comprehensive understanding of CGR, and as the baseline GENRE* is our own implementation, we wanted to share the official results of DPR, SEAL, and GENRE to show the reliability of our experimental results. As the objective of the paper was to show that changing the decoder vocab embeddings space to contextualized embedding space is a simple but effective method, the table mostly focused on how much the performance of CGR increased from GENRE* and how the performance of different architectures of CGR would differ.
>
> > In Table 2, GENRE, DSI, and SEAL’s performance are so low on NQ-320K. I would appreciate it if the authors could clarify the setting (sorry if I missed it) or maybe retrain the GENRE on NQ-320K in the same setting as CGR (it shouldn’t be difficult since the code should be similar to CGR).
>
> In Table2, the performances of GENRE and SEAL are from the official SEAL paper [1], and the performance of DSI is from the official DSI paper [2] (caption of Table 2). As the exact dataset of NQ-320k is not open-sourced, we constructed the dataset ourselves with the official NQ dataset. We have checked the statistics of the dataset is the same as in [2].
>
> As mentioned in the “Common response to reviewers” comment, the performance of SEAL from the official paper seems to be underestimated, and the authors of SEAL are working on resolving the issue (https://github.com/facebookresearch/SEAL/issues/14). Therefore, we have removed the SEAL performance. Also, we have replaced the performance of GENRE (Bart-large) with GENRE* (T5-large) and have added the performance of BM25 and CGR_contra.
>
> We have analyzed why CGR shows higher performance than GENRE* in NQ-320k compared to KILT NQ. We hypothesize that the performance gap between the two datasets comes from how much lexical overlap exists between the query and the retrieval sequence for the two datasets; KILT NQ shows a higher lexical overlap between the query and the retrieval sequence compared to NQ-320k. When comparing the performance of BM25 over other models in NQ-320k (Table 2) and KILT NQ (Table 1), we have seen that BM25 shows relatively lower performance in NQ-320k than KILT NQ. (As BM25 is a sparse model, we can assume that when the performance is low, there would be less lexical overlap between the query and the retrieval sequence.) Also, when comparing the first 20 examples from each dataset manually, we have seen that KILT NQ tends to have a higher overlap between the query and the retrieval target compared to that of NQ-320k. As shown in the last paragraph of Section 5 and Appendix C.6, GENRE(*) tends to depend highly on word overlap whereas CGR tends to show strong performance in both cases with high and low word overlap as CGR can utilize the content information in the contextualized vocab embeddings (external memory).
>
> ### Recommendation
> > The authors could improve the clarity — e.g., the authors can make it clearer what text they put into the encoders for the contextual embeddings (in the paper they only vaguely mention that they are titles).
>
> We have updated the caption of Figure 1 of the paper following your recommendations. Thank you for the suggestion!
>
> Also, please note that CGR is not limited to cases where the retrieval target is a title but can also be generalized to various target sequences as mentioned in Section 3. To show such cases, we have added the result of CGR_base in NQ-320k when the retrieval target is not the title but a random document ID. Please refer to Appendix C.8 of the updated paper for more details.

---

> > ### Author Response · Authors · 2022-11-15
> > **Response to Reviewer DJS8**
> >
> > > Figure 1 is a bit confusing to me (especially Cape(1), Cape(2)).
> >
> > We have updated Section 3 of the paper following your recommendations. Thank you for the suggestion!
> >
> >
> > ### References
> > [1] Autoregressive Search Engines: Generating Substrings as Document Identifiers (https://arxiv.org/abs/2204.10628)
> >
> > [2] Transformer Memory as a Differentiable Search Index (https://arxiv.org/abs/2202.06991)

---

> > ### Comment · Reviewer_DJS8 · 2022-11-16
> > **Review update**
> >
> > Thanks for the response! I realized that a previous comment of mine was not made public. Sorry about that! I paste it here:
> >
> > > I realize I made a mistake in the review. The authors did compare to a reproduced GENRE baseline. However, there still exist other flaws as mentioned in the review. Besides, to help us better understand the effect of the proposed model, the authors should conduct more ablation study, e.g., how constrained decoding/beam width/pre-trained models affect the performance of GENRE and CGR? I also have another question: did the authors use T5 1.0 or 1.1? Those two models are very different since 1.0 are trained with supervised data.
> >
> > No need to answer those questions since they are so last minute. I just put them here in case they would be helpful.

---

> > > ### Author Response · Authors · 2022-11-18
> > > **Response to Review update of DJS8**
> > >
> > > Hello Reviewer DJS8,
> > >
> > > Thank you for your valuable time in reviewing our paper. We appreciate your detailed comments about our paper.
> > >
> > > > 1. The authors did compare to a reproduced GENRE baseline. However, there still exist other flaws as mentioned in the review.
> > >
> > > In the previous review, the reviewer mentioned:
> > >    * adding more baselines in Table 1
> > >       * We added more baselines (DPR and BM25 in the same setting as CGR) in Table 1, and the results of the dev set in Table 10.
> > >    * matching the pre-trained model of baselines with the pre-trained model of CGR (T5-large)
> > >       * In the previous rebuttal, we clarified that for Table 1, 3, 4, and 5 the baseline model GENRE uses the same pretrained model as CGR.
> > >       * For Table 2, we replaced the performance of GENRE to GENRE with T5-large as the base model. We also removed the performance of SEAL in Table 2, as we noticed from the Github issue of SEAL that the number may be underestimated.
> > >       * Therefore, now all results of GENRE and CGR have the same pre-trained model T5-large.
> > > Thereby we believe that we have answered all the questions from the reviewer in the previous rebuttal.
> > > It would be very helpful if the reviewer could mention any parts we missed or could clarify the existing flaws mentioned in the review. Thanks!
> > >
> > > > 2. How constrained decoding/beam width/pretrained models affect the performance of GENRE and CGR
> > >
> > > We have experimented with the model in several different settings with a toy set which we would like to share.
> > >
> > >    * Constrained decoding: as constrained decoding is necessary to ensure that all the generated (retrieved) sequences are in the corpus, we did not perform without the prefix tree. However, we agree that analyzing the difference between the case with and without the prefix tree would be interesting.
> > >    * Beam width: We also experimented with beam size 5 and could see that beam size 10 shows higher performance, but the difference was marginal. We used beam size 10 to match the official GENRE setting.
> > >    * Pretrained models: We have trained the model with BART-large in a toy set but did not see a huge difference in the performance.
> > >
> > > > 3. Did the authors use T5 1.0 or 1.1?
> > >
> > > We trained all models with T5 1.0.
> > >
> > > * References
> > >
> > > [1] Autoregressive Entity Retrieval (https://arxiv.org/abs/2010.00904)
> > >
> > > [2] Generative Multi-hop Retrieval (https://arxiv.org/abs/2204.13596)

---

> > ### Comment · Reviewer_DJS8 · 2022-11-16
> > **Followup questions**
> >
> > Thanks for the solid and comprehensive response! It answered a lof of my questions about the paper. I have some followup questions but you don't need to answer those that require new experiments given the limited time:
> >
> > (1) Would it be also possible to add DPR (trained on NQ and TriviaQA) to the KILT results? The main reason I ask for that is that in Table 1 DPR is much better than GENRE, but in the original GENRE paper they claim GENRE surpassed DPR. I assume it might also be relevant to the data that the original GENRE model used.
> >
> > (2) What is the encoder for the reproduced DPR?
> >
> > (3) In Table 3 the title only version is very close to CGR and much better than GENRE. It seems to suggest that CGR does not work because the contextual embedding incorporates the passage information. Do you have any comments on that?
> >
> > (4) It seems that DPR is similar to CGR on NQ but much worse on TriviaQA. Any comment?
> >
> > Thanks for the effort that you put into on reproducing the baselines. I believe this will become a solid work in the end no matter what the final review result is

---

> > > ### Author Response · Authors · 2022-11-18
> > > **Response to Followup Questions of DJS8**
> > >
> > > Thank you for your insightful comments.
> > >
> > > >(1) Would it be also possible to add DPR (trained on NQ and TriviaQA) to the KILT results? The main reason I ask for that is that in Table 1 DPR is much better than GENRE, but in the original GENRE paper they claim GENRE surpassed DPR. I assume it might also be relevant to the data that the original GENRE model used.
> > >
> > > >(4) It seems that DPR is similar to CGR on NQ but much worse on TriviaQA. Any comment?
> > >
> > > When analyzing the results of CGR, DPR, and GENRE (along with other generative retrieval models) from the official results and our results, we have found two interesting features:
> > >
> > > 1. Having a large training dataset is important for high performance in the generative retrieval model
> > > 2. The performance of the generative retrieval models tends to align with the performance of the sparse model, in which the high performance of the sparse model indicates that there is a high word overlap between the input (query) and the retrieval sequence (output).
> > >
> > > We think such observation answers question 1 and 4. Below we add details of how the observation relates to the questions.
> > >
> > > >Q. Why is DPR much better than GENRE on NQ in Table 1 (which is not aligned with the results of the original GENRE paper)
> > >
> > > A. We think of the result in two aspects. One is that the number of datasets is different, leading to different results; the result in Table 1 is GENRE* trained in a single dataset (NQ only, 0.7% of KILT+BLINK), and the result in GENRE paper is the result trained in all KILT plus the BLINK dataset. From observation 1, we can assume that as the number of training datasets of NQ is about 0.7% of the KILT + BLINK dataset, it made the performance of GENRE* lower than GENRE trained on all. The other is that sparse model show relatively low performance in NQ. Table 3 of [1] shows the performance of DPR, tf-idf, and GENRE. The performance of tf-idf in NQ and TQA are 28.1 and 46.4, respectively, and the increment of GENRE over DPR is 111% and 156%, respectively, showing that even GENRE trained on full dataset shows relatively low-performance gain in NQ. Also, please note that DPR in [1] is trained on official NQ, TQA, WQ, and TREC and inferenced over KILT NQ and TQA, which we hypothesize that the performance will increase when it is trained on KILT NQ and TQA.
> > >
> > > >Q. It seems that DPR is similar to CGR on NQ but much worse on TriviaQA. Any comment?
> > >
> > > A. We think the different trend between NQ and TQA comes from the characteristic of each dataset; how much word overlap exists between the two datasets (Observation 2). As TQA shows high performance in both tf-idf of the GENRE official paper and the bm25 score of our paper, we can infer that the dataset has a higher overlap which may lead to higher performance of generative retrieval models. Table 10 of the updated paper shows that not only GENRE* and CGR but also GENRE (official GENRE) and SEAL show much higher performance in TQA compared to NQ.
> > >
> > > Here are some references to where we could find some observations
> > >
> > > 1. Having a large training dataset is important for higher performance in the generative retrieval model
> > >    * [1] has mentioned that using all datasets increases the performance of GENRE.
> > >    * NCI and GMR have shown that the performance increases by adding more training datasets by the pseudo-query generation method.DSI and SEAL also use a large number of training datasets by additionally using unsupervised datasets.
> > >    * Table 1 of our paper shows that both GENRE(*) and CGR tends to show higher performance when using more datasets.
> > > 2. The performance of the generative retrieval models tends to align with the performance of the sparse model, in which the high performance of the sparse model indicates that there is a high word overlap between the input (query) and the retrieval sequence (output).
> > >    * Table 3 of [1] show that there is some correlation between tf-idf and the performance gap between GENRE and DPR in NQ and TQA (as DPR has not seen the other datasets).
> > >    * The last paragraph of Section 5 of CGR shows that word overlap affects the performance of generative retrieval. Results of Table 1 and Table 2 also show that there is some correlation between the performance of the sparse model and the generative retrieval model.
> > >
> > > > (2) What is the encoder for the reproduced DPR?
> > >
> > > Following the official setup of the DPR code, the encoder of the reproduced DPR uses the BERT base model.

---

> > > > ### Author Response · Authors · 2022-11-18
> > > > **Response to Followup Questions of DJS8**
> > > >
> > > > > (3) In Table 3 the title only version is very close to CGR and much better than GENRE. It seems to suggest that CGR does not work because the contextual embedding incorporates the passage information. Do you have any comments on that?
> > > >
> > > > The CGR of title only version is closer to CGR than GENRE in the sense that it also uses contextualized vocab embeddings (external memory) where the context is from the title and contains multiple contextualized embeddings for each token in the title. Therefore, we think the result that the CGR-title-only version is close to CGR than GENRE does not imply that “CGR does not work because the contextual embedding incorporates the passage information”.
> > > >
> > > > For more details, we would like to highlight the differences between CGR-title-only with CGR and GENRE.
> > > >    * Differences between CGR and CGR-title-only
> > > >       * CGR uses not only the title but also the passage as the context; CGR contains much more information in the context. Therefore in the examples in Table 5, we could see that CGR-title-only fails to retrieve the first five examples, which need the passage information in the contextualized embeddings to successfully retrieve the sequence.
> > > >    * Difference between GENRE and CGR-title-only
> > > >       * CGR-title-only uses the contextualized vocab embeddings constructed with the title information as the context. Although the context information is much simple compared to the CGR, with also uses the document information, we could see that such title information especially helps in catching the “detailed” information. Previous work ((3) in Appendix C.2.1 of [2]) showed that as the generative retrieval model generates (retrieves) the sequence in a uni-directional way, it misses some details information (e.g., who’s the singer, whether it is a novel or a film, which country it is). In the last example of Table 5, we can see that CGR-title-only catches the correct sequence better than GENRE, which leads to higher performance.
> > > >       * Using multiple token embeddings for a single token is also one of the important novelties of CGR architecture (Contextualized Embedding Space of Figure 1 shows the example). As the model has more than one representative embedding for a token, it can contain more diverse and specified information for the embeddings. We hypothesize that the method also contributed to the performance gap between GENRE and CGR-title-only.

---

### Author Response · Authors · 2022-11-15
**Common response to reviewers**

We thank all reviewers for their valuable comments. We have individually responded to each reviewer’s comment and have updated the paper. In this general response, we put the details of our objective of the paper, results of additional experiments, and how CGR differs from kNN-LM.

### 1. A brief overview of our objective of the work
To the best of our knowledge, CGR is the first work to entirely replace the decoder vocab space to contextualized embedding space from the training step. Therefore, most of our works focused on presenting various architectures of CGR (CGR_base, CGR_async, and CGR_contra) to comprehensive understanding on how the relationship between EMB (encoder model that extracts the contextualized embeddings) and RET (the retrieval model trained on retrieval task with contextualized embeddings as decoder vocab embeddings) affects the performance. Also, we focused on presenting various benefits of using contextualized vocab embeddings (external memory with context information encoded) over vanilla vocab embeddings when retrieving the target sequence in a generative way. Note that we used the contextualized embeddings both during the training and inference time. Although it is a simple method of changing the decoder vocab space during the training step, we could find that by using the encoded information inside the contextualized embeddings, there are various benefits:
1. higher retrieval performance (Table1, Table2)
2. better zero-shot ability (Table4)
3. overcomes the dependency on lexical overlap between the query and the retrieval target (Table 5, Appendix C.6).


### 2. Additional experiments during the rebuttal period
During the rebuttal period, we have added additional results in Table 1 and Table 2. Please note that our direct baseline is GENRE* (GENRE* is GENRE reproduced in the same setting as ours, using T5-large as the base model) as our objective is to present a simple method of changing the decoder vocab space from vanilla model vocab embeddings to contextualized vocab embeddings during the training time and show that it helps as the model can use the encoded contextualized information during the decoding step. The results of other models are presented for a more comprehensive understanding of CGR.
To address the concern of rcjJ and DJS8, we have added more baselines (BM25 and DPR) in Table 1. In Table 2, we have replaced the result of GENRE to GENRE* to match the base model and removed the result of SEAL as it may have been underestimated and the authors are working on resolving the issue (https://github.com/facebookresearch/SEAL/issues/14). We have added the result of GENRE* (GENRE based on T5-large), BM25, and CGR_contra to keep the setting similar. We have also added KILT dev results in Appendix C.7.
To address the concern of UYMW, we have added the performance of CGR_base in NQ-320k when the retrieval target is document ID as in DSI in Appendix C.8. In this case, as the retrieval target is document ID (not the title), the direct baseline is DSI (not GENRE*). From the results in Table 11, we have seen that CGR is generalizable to various forms of retrieval targets as mentioned in Section 3.

---

> ### Author Response · Authors · 2022-11-15
> **Common response to reviewers**
>
> ### 3. How CGR differs from kNN-LM
> Although there are several previous works using contextualized embeddings such as the kNN-LM model as mentioned in the Related Work section, CGR is different from kNN-LM in many aspects as shown in the table below.
>
> |  | kNN-LM | CGR |
> |---|---|---|
> | Task | language modeling | Retrieval |
> | The objective of using contextualized embeddings | finding the most plausible next token by finding the token with a similar context  till the previous token from the training corpus | encoding the nearby context to the representative words (e.g., titles, document ID)  so that the token embeddings itself will contain the overall information of the  document without the need to generate the whole document |
> | Are the contextualized embeddings  used during the training step? | No | Yes |
> | Model to extract contextualized embeddings  (** we will name the model as  EMB in the  table following the naming in CGR) | decoder-only model (uni-directional) | encoder of the encoder-decoder model (bi-directional) |
> | Is EMB the same model as the model used during  training for the objective? | Yes | No (There are two submodules EMB, the model that extracts the contextualized embeddings,  and RET, the model used during training for the objective.) |
> | How is EMB trained before extracting  contextualized embeddings? | language modeling task, as it uses the same model for language modeling  and extracting contextualized embeddings | For CGR_contra, EMB is trained with contrastive loss.  For CGR_base, it uses a frozen pre-trained model. |
> | Context used to extract contextualized embeddings  of t-th token from EMB (saved in datastore) | tokens at i=0, 1, .., t-1 (previous tokens) | tokens at i=0, 1, .., t-1, t, t+1, …, T (all tokens in the context) |
> | Which embeddings from EMB is used for contextualized embedding? | input embedding of the last FFN layer when given the previous tokens as input. | output embeddings of EMB |

---

### Decision · Program_Chairs · 2023-01-20

**Decision:**

Reject

**Justification For Why Not Higher Score:**

Not good enough

**Justification For Why Not Lower Score:**

N/A

**Metareview: Summary, Strengths And Weaknesses:**

This paper proposes a semi-parametric approach to generation based retrieval. While generating a target sequence, the decoder accesses a memory bank of pre-computed token embeddings. Consider that similar techniques have been extensively studied in language modeling (KNN-LM, Yogatama et al), translation (Khandelwal et al, ICLR'21) and other generation applications including dialog (Fan et al), the novelty of the approach is somewhat limited. Also, reviewers had further concerns about the experiments (DJS8, rcjJ), presentation/writing (rcjJ). Considering all these, I am in favour of rejecting the paper in its current form.

**Summary Of Ac-Reviewer Meeting:**

N/A